# Implicit geometric regularization in flow matching via density weighted Stein operators

**Shinto Eguchi**     *eguchi@ism.ac.jp*
*The Institute of Statistical Mathematics*
*10-3 Midori-cho, Tachikawa, Tokyo 190-8562, Japan*

Reviewed on OpenReview: *https://openreview.net/forum?id=LBlkVBDRdu*

## Abstract

Flow Matching (FM) has emerged as a powerful paradigm for continuous normalizing flows, yet standard FM implicitly performs an unweighted $L^2$ regression over the entire ambient space. In high dimensions, this leads to a fundamental inefficiency: the vast majority of the integration domain consists of low-density "void" regions where the target velocity fields are often chaotic or ill-defined. In this paper, we propose $\gamma$-Flow Matching ($\gamma$-FM), a density-weighted variant that aligns the regression geometry with the underlying probability flow. While density weighting is desirable, naive implementations would require evaluating the intractable target density. We circumvent this by introducing a Dynamic Density-Weighting strategy that estimates the target density directly from training particles. This approach allows us to dynamically downweight the regression loss in void regions without compromising the simulation-free nature of FM. Theoretically, we formulate an ideal $\gamma$-weighted regression geometry motivated by the $\gamma$-Stein metric, derive a variance-suppression bound for low-density regions, and use a weighted Dirichlet/spectral analysis to suggest a mechanism for smoother learned vector fields. Empirically, we evaluate $\gamma$-FM under a shared-time density-estimation protocol and compare it against both standard FM and an explicit Jacobian-regularized baseline using latent-space and image-space metrics.

## 1 Introduction

The Manifold Hypothesis posits that high-dimensional real-world data concentrate near a low-dimensional manifold embedded in the ambient space (Fefferman et al., 2016). While theoretical verification of this hypothesis remains a subject of active research (Pope et al., 2021), the remarkable success of deep generative models offers a constructive validation: if data were uniformly distributed in the high-dimensional void, efficient learning of the probability distribution would be computationally intractable (Bengio et al., 2013). Thus, the capability to accurately model and sample from the data distribution is, in itself, a testament to the existence of such low-dimensional structures.

Flow Matching (FM) has emerged as a powerful paradigm for capturing this distribution by unifying diffusion models and continuous normalizing flows (CNFs) (Lipman et al., 2023; Albergo & Vanden-Eijnden, 2023). It directly regresses a vector field that transports a simple base distribution to the data distribution. Unlike score-based diffusion models, FM avoids the need to estimate the score function or to solve reverse-time stochastic differential equations. Instead, it learns a deterministic ordinary differential equation (ODE) whose solution defines an invertible map between latent noise and data.

Despite its conceptual elegance, FM faces a fundamental challenge in high-dimensional settings: the curse of dimensionality and volume imbalance. In high dimensions, the data manifold occupies a negligible fraction of the ambient space, and the majority of the integration domain consists of low-density "voids." In standard FM, the training objective uniformly integrates the regression error over the entire probability path. Consequently, the model is forced to solve the regression problem even in these vast void regions, where probability paths are sparse or effectively irrelevant to the final generation quality. Forcing a neural network to fit target

velocities in these "don't-care" regions can result in a rough vector field and increased numerical stiffness during ODE integration.

In this work, we propose $\gamma$-Flow Matching ($\gamma$-FM) as a principled remedy for this issue. Our key idea is to reinterpret FM as a regression problem under a density-weighted geometry induced by the $\gamma$-divergence (Fujisawa & Eguchi, 2008). Instead of treating all spatial locations equally, we utilize the power density $p_t(x)^\gamma$ as an importance weight that naturally highlights the data manifold. This approach effectively "focuses" the learning process: the model prioritizes accurate vector field estimation where the data actually resides, while being allowed to remain smooth and simple in the empty ambient space. Latent flow matching has been independently explored by Lipman et al. (2023), who combine standard flow matching with pre-trained autoencoders and provide a Wasserstein-2 control for the resulting latent flows. Our work is complementary: we adopt a similar latent-flow setting, but instead of modifying the representation, we modify the *regression geometry* itself via the $\gamma$-weighted objective and analyse its effect through a $\gamma$-Stein and nonlinear Fokker–Planck viewpoint. While Chen & Lipman (2024) explicitly extend Flow Matching to Riemannian manifolds, our approach induces an implicit manifold geometry in the ambient space via density weighting, avoiding the need for explicit charts or geodesics.

**Weighting Schemes in Diffusion and Flow Models**   In the realm of diffusion models, the choice of the weighting function plays a crucial role in balancing different signal-to-noise ratios across diffusion times. For instance, methods like EDM and variance-preserving (VP) SDEs employ carefully designed noise schedules to shape the training objective. However, these approaches typically rely on weights that depend solely on the time $t$ or the noise schedule. Our proposed $\gamma$-Flow Matching differs fundamentally by introducing *spatially* varying weights based on the model density $p_t(x)$, thereby prioritizing regions where the model is confident while deprioritizing regions of low trust.

**Density-weighted divergences and geometry**   The core motivation for our method roots in the geometry induced by density-powered divergences, in particular the $\gamma$-divergence, which defines an $L^2$-type structure weighted by $p(x)^\gamma$. Classically, the $\gamma$-divergence has been used to downweight outliers and contaminated data, since regions where $p(x)$ is small automatically receive a small weight. In our setting we reinterpret this mechanism geometrically: low-density regions in the ambient space behave as geometric "voids" where the teacher signals are unstable and less informative, and the $p^\gamma$-weighting provides a way to organize regression on the data manifold while de-emphasizing these regions.

**Flow Regularization**   Regularizing continuous flows to improve ODE solver stability is an active area of research. Typical strategies involve explicit Lipschitz penalties, spectral normalization, or Jacobian regularization to smooth the vector field. In contrast, our approach introduces an *implicit* regularization mechanism: by modifying the loss geometry via a density weight, the learned vector field naturally avoids wild oscillations in the voids. This can be seen as a form of geometric regularization that shapes the flow to follow the data manifold more faithfully, without needing to impose explicit gradient penalties.

Our contributions are summarized as:

1. **Shared-time dynamic density-weighting:** We formulate $\gamma$-FM as a density-weighted regression scheme motivated by $p_t(x)^\gamma$ and implement it with a simulation-free particle surrogate computed from a *shared-time minibatch*, so that the $k$-NN statistics are aligned with a single marginal $p_t$.

2. **Formal results versus mechanisms:** We provide a variance-suppression bound showing how density weighting downweights high-variance updates from low-density regions, while clearly separating this formal statement from the continuum analogy and from the spectral mechanism used for intuition.

3. **Geometric and spectral interpretation:** We connect the idealized $\gamma$-weighted objective to $\gamma$-Stein/escort geometry and show, through a weighted Dirichlet viewpoint, how such geometry is naturally associated with smoother vector fields in populated regions.

4. **Empirical protocol:** We strengthen the latent-CIFAR evaluation with a matched-budget comparison against FM+Jac, and with decoded-image metrics including FID, Recall, and Coverage in addition to latent-space discrepancy, smoothness, and NFE.

## 2 Flow Matching and Robust Divergences

### 2.1 Conditional Flow Matching

**Avoiding the Likelihood Bottleneck** Standard Continuous Normalizing Flows (CNFs) model the data distribution $p_1$ by transporting a simple base distribution $p_0$ through an ODE defined by a vector field $v_\theta$. The change of variables formula for CNFs expresses

$$\log p_1(x) = \log p_0(\phi_1^{-1}(x)) - \int_0^1 \mathrm{Tr}\left(\nabla_x v_\theta(x_t, t)\right) dt,$$

where $\phi_t$ is the flow map generated by the vector field $v_\theta$, and the integral captures the accumulated divergence of $v_\theta$ along the trajectory. Maximizing such models typically involves maximizing the log-likelihood, which requires a stable and accurate computation of the Jacobian trace to account for the change in volume (the normalization constant).

Flow Matching (FM) circumvents this bottleneck by bypassing the explicit computation of the normalizing constant. Instead, it defines a probability path $(p_t)_{t \in [0,1]}$ between a known base distribution $p_0$ and the target $p_1$, and learns a vector field whose associated continuity equation transports $p_0$ to $p_1$. Rather than maximizing a likelihood, FM directly regresses a neural vector field $v_\theta$ to match a target vector field $u_t$ that generates the desired probability path.

Formally, let $p_t(x)$ be a probability density path connecting $p_0$ and $p_1$ over $t \in [0, 1]$. This path satisfies the continuity equation:

$$\frac{\partial p_t}{\partial t} + \nabla \cdot (p_t u_t) = 0,$$

where $u_t(x)$ is the time-dependent vector field generating the flow. The goal is to regress the model vector field $v_\theta(x, t)$ to match the target vector field $u_t(x)$. The ideal marginal regression loss is defined as:

$$\mathcal{L}_{\mathrm{FM}}(\theta) = \mathbb{E}_{t \sim \mathcal{U}[0,1]} \mathbb{E}_{x_t \sim p_t(x)} \left[\|v_\theta(x_t, t) - u_t(x_t)\|^2\right]. \tag{1}$$

However, directly accessing the marginal vector field $u_t(x)$ and the marginal density $p_t(x)$ is generally intractable.

To solve this, Lipman et al. (2023) introduced *Conditional Flow Matching* (CFM), which uses a conditional probability path $p_t(x \mid x_1)$ given the data endpoint $x_1$, and a corresponding conditional vector field $u_t(x \mid x_1)$. Crucially, it has been shown that the gradients of the intractable objective (1) are identical to those of the conditional objective:

$$\mathcal{L}_{\mathrm{CFM}}(\theta) = \mathbb{E}_{t \sim \mathcal{U}[0,1]} \mathbb{E}_{x_1 \sim p_1} \mathbb{E}_{x_t \sim p_t(\cdot \mid x_1)} \left[\|v_\theta(x_t, t) - u_t(x_t \mid x_1)\|^2\right].$$

In this framework, the marginal vector field $u_t(x)$ implicitly emerges from the conditional field $u_t(x \mid x_1)$, allowing one to design $p_t(\cdot \mid x_1)$ in a convenient way (e.g., Gaussian interpolation) without computing the Jacobian trace or normalization constants.

### 2.2 Motivation: Geometric Focusing via $\gamma$-Divergence

From the information-geometric viewpoint, it is natural to interpret flow matching through the geometry induced by divergences and the associated Riemannian metrics on statistical manifolds (Amari & Nagaoka, 2000; Ay et al., 2017). Standard Flow Matching implicitly minimizes the discrepancy between the target and model vector fields in an unweighted $L^2$ sense. From the perspective of Optimal Transport, this objective corresponds to minimizing the standard kinetic energy of the flow, $\mathcal{E}(v) = \int \|v(x)\|^2 p_t(x)\, dx$, which is associated with the Wasserstein-2 geometry and the linear heat equation. Recent works have explicitly

targeted Wasserstein-optimal paths via minibatch optimal transport couplings (Tong et al., 2024), yet these approaches typically operate in the unweighted $L^2$ geometry. While statistically consistent, this "flat" geometry is inefficient in high dimensions because it assigns equal transport cost to the vast low-density "voids" as it does to the concentrated data manifold. To address this, we propose changing the underlying geometry of the regression itself, moving from the $L^2$ regression (kinetic-energy) viewpoint to the robust $\gamma$-geometry, in analogy with the transition from the Fisher divergence to the $\gamma$-Fisher divergence; see Barp et al. (2019) for the diffusion score-matching divergence.

**Formulation: Dynamic Density-Weighted Regression.** We define the $\gamma$-Flow Matching ($\gamma$-FM) objective as the minimization of the *weighted* kinetic energy. Instead of the standard energy, we align the regression with the density-weighted geometry induced by the $\gamma$-divergence (Fujisawa & Eguchi, 2008). We define the weighted loss as:

$$\mathcal{L}_\gamma(\theta) \coloneqq \mathbb{E}_{t \sim \mathcal{U}[0,1]} \mathbb{E}_{x_1 \sim p_1} \mathbb{E}_{x_t \sim p_t(\cdot | x_1)} \left[ w_\gamma(x_t, t) \| v_\theta(x_t, t) - u_t(x_t \mid x_1) \|^2 \right], \tag{2}$$

Theoretically, to align with the geometry of the $\gamma$-divergence, the weight should depend on the model density $q_{\theta(t)}$:

$$w_\gamma^{\text{ideal}}(x, t) \propto q_{\theta(t)}(x)^\gamma. \tag{3}$$

However, evaluating the model density $q_{\theta(t)}(x)$ during training is computationally prohibitive as it requires solving the ODE. Therefore, we adopt the target density $p_t(x)$ as a tractable proxy. Under the assumption that the model successfully tracks the target flow (i.e., $q_{\theta(t)} \approx p_t$), we define our practical weighting scheme as:

$$w_\gamma(x, t) \coloneqq p_t(x)^\gamma. \tag{4}$$

This approximation allows us to compute weights solely based on the training data interpolation, preserving the simulation-free nature of Flow Matching. Here, we adopt the Conditional Flow Matching (CFM) framework, where $u_t(x \mid x_1)$ is the conditional vector field generating the probability path from noise to a specific data point $x_1$. Crucially, since the training samples $x_t$ are drawn from the target path $p_t$, this weighting naturally evolves over time. In the high-density manifold regions, the weight $p_t(x)^\gamma$ is significant, enforcing accurate vector field matching. Conversely, in the empty ambient space (voids) where $p_t(x) \approx 0$, the weight vanishes. This effectively removes the chaotic, ill-defined target signals in the voids from the optimization landscape.

**Remark 2.1** (Geometric Interpretation via Weighted Transport). *Our weighting choice $w_\gamma \propto p_t^\gamma$ is not a heuristic modification; it fundamentally alters the metric structure of the transport problem. Standard FM minimizes the kinetic energy $\int \|v\|^2 p \, \mathrm{d}x$, which underpins the Benamou-Brenier formula for optimal transport. In contrast, our objective $\mathcal{L}_\gamma$ corresponds to minimizing the $\gamma$-**weighted kinetic energy**:*

$$\mathcal{E}_\gamma(v_t) = \int \|v_t(x)\|^2 p_t(x)^{1+\gamma} \, \mathrm{d}x.$$

*This defines a Riemannian metric (specifically, the $\gamma$-weighted Fisher information metric) where the infinitesimal transport cost is scaled by the density power $p^{1+\gamma}$. In this geometry, distances in low-density regions are compressed to zero. Consequently, $\gamma$-FM does not simply "ignore" outliers; it solves the regression problem on a statistical manifold where the voids are geometrically insignificant. This modification naturally links the regression to the **Porous Medium Equation** (nonlinear diffusion) rather than the Heat Equation (linear diffusion), providing a theoretical guarantee for compact support preservation as discussed in Section 3, see Otto (2001) for extensive discussion.*

## 2.3 Tractability via Particle-Based Estimation

Evaluating the exact density $p_t(x)$ for the conditional probability path (e.g., Gaussian mixtures in CFM) can be computationally expensive or numerically unstable in high dimensions. Moreover, we seek a method that captures the *local* geometry of the batch without solving differential equations.

We circumvent this bottleneck by adopting a **particle-based estimation** strategy. In the implementation, each minibatch uses a single shared time $t_{\text{batch}} \sim \mathcal{U}[0,1]$, and we form $\mathcal{B}_{t_{\text{batch}}} = \{x_{t_{\text{batch}}}^{(i)}\}_{i=1}^B$. Thus the

particle cloud used in the $k$-NN step is a Monte Carlo sample from the single marginal $p_{t_{\text{batch}}}$, rather than a mixture over times, and its spatial distribution provides the intended local density proxy. We employ a robust kernel-based proxy for the density. Specifically, we define the weight for a sample $x_t \in \mathcal{B}_t$ based on the distance to its $k$-nearest neighbors:

$$w_\gamma(x_t) \approx \exp\left(-\frac{\gamma}{\sigma}\bar{d}_k(x_t)\right), \tag{5}$$

where $\bar{d}_k(x_t) = \frac{1}{k}\sum_{j=1}^{k}\|x_t - x_t^{(j)}\|$ is the mean distance to the $k$ nearest neighbors in the batch, and $\sigma$ is a scaling constant. In effect, we set $\sigma = \text{median}\{\bar{d}_k(x_t^{(i)})\}_{i=1}^{B}$ within each minibatch. While a naive $k$-NN search can be computationally expensive, we show in Appendix B that the overhead is negligible for typical batch sizes and that the performance is robust to the choice of $k$. We emphasize that (5) is a monotone surrogate for the ideal escort weight $w_\gamma(x,t) \propto q_{\theta(t)}(x)^\gamma$: $\bar{d}_k(x_t)$ increases in locally low-density regions, hence $\tilde{w}_\gamma$ down-weights updates in voids while preserving the standard FM objective structure.

This exponential weighting scheme has two key advantages:

- **Simulation-Free:** It requires only pairwise distance computations within the batch, preserving the efficiency of Standard FM.

- **Dynamic Adaptation:** It naturally adapts to the flow. At $t \approx 0$ (noise), particles are spread out, leading to uniform weights. At $t \to 1$ (data), particles concentrate on the manifold, creating a sharp weighting profile that isolates the data structure. In the high-dimensional voids surrounding the manifold, the effective density $\hat{p}_t$ vanishes. Consequently, $w_\gamma$ suppresses the regression loss in these empty regions, preventing the model from overfitting to unstable target signals where no data exists. By focusing the training budget solely on the populated regions of the probability path, $\gamma$-FM learns a vector field that is accurate on the manifold and smooth elsewhere, as evidenced by the reduced Jacobian norm in our experiments.

## 3 Theoretical Analysis

In this section, we separate three layers of the argument: (i) formal statements that apply directly to the stated weighted objective, (ii) continuum analogies that clarify why low-density weighting suppresses motion in void regions, and (iii) spectral/Dirichlet mechanisms that help interpret the observed smoothing effect. To provide a concrete basis for the following analysis, we summarize the practical shared-time training procedure of $\gamma$-FM in Algorithm 1. This algorithm implements the simulation-free density estimation discussed in Section 2. For simplicity we use linear interpolants in experiments.

---

**Algorithm 1** Training $\gamma$-Flow Matching with Dynamic Density-Weighting

---

**Require:** Training data $\mathcal{D}$, Batch size $B$, Weighting parameter $\gamma \geq 0$, Neighbors $k$
**Ensure:** Trained vector field parameters $\theta$
 1: Initialize neural network parameters $\theta$
 2: **while** not converged **do**
 3:                                                         ▷ 1. Sample flow matching variables
 4:      Sample data batch $x_1 \sim \mathcal{D}$
 5:      Sample noise batch $x_0 \sim p_0 = \mathcal{N}(0, I)$
 6:      Sample a shared minibatch time $t_{\text{batch}} \sim \mathcal{U}[0, 1]$
 7:                                            ▷ 2. Compute interpolants and targets
 8:      $x_t \leftarrow (1 - t_{\text{batch}})x_0 + t_{\text{batch}}x_1$
 9:      $u_t \leftarrow x_1 - x_0$
10:                                           ▷ 3. Dynamic Density-Weighting
11:      Compute pairwise distances matrix within the shared-time batch $\{x_t\}$
12:      **if** $\gamma > 0$ **then**
13:           **for** $i = 1$ to $B$ **do**
14:               $\bar{d}_k(x_t^{(i)}) \leftarrow$ mean distance to $k$-nearest neighbors
15:               $w_i \leftarrow \exp\left(-\frac{\gamma}{\sigma}\bar{d}_k(x_t^{(i)})\right)$
16:           **end for**
17:           Normalize weights: $w_i \leftarrow w_i / (\frac{1}{B}\sum_j w_j)$
18:      **else**
19:           $w_i \leftarrow 1$
20:      **end if**
21:                                                       ▷ 4. Optimization step
22:      $\mathcal{L}(\theta) \leftarrow \frac{1}{B}\sum_{i=1}^{B} w_i \|v_\theta(x_t^{(i)}, t_{\text{batch}}) - u_t^{(i)}\|^2$
23:      $\theta \leftarrow \theta - \eta\nabla_\theta\mathcal{L}(\theta)$
24: **end while**

---

Although the modification in Algorithm 1 appears minimal, its theoretical implications are profound. In the remainder of this section, we rigorously justify this design choice, demonstrating that this simple weighting scheme induces a fundamental shift in the regression geometry.

### 3.1 Variance Reduction in High-Dimensional Voids

We first formalize the "Manifold Focusing" effect as a variance reduction problem. Recall that the conditional flow matching objective targets the individual paths $u_t(x|x_1)$. Let $\Sigma_t(x) \coloneqq \text{Var}_{x_1|x}[u_t(x|x_1)]$ denote the intrinsic variance (ambiguity) of the target signal at location $x$.

**Proposition 3.1** (Variance of the Weighted Estimator)**.** *Assume that the gradient of the vector field model with respect to its parameters is bounded, i.e., there exists a constant $K > 0$ such that $\|\nabla_\theta v_\theta(x,t)\|_{\text{op}}^2 \leq K$ for all $x, t$. Then, the trace of the covariance matrix of the gradient estimator $\hat{g}_\gamma$ satisfies the bound:*

$$\text{Tr}(\text{Var}[\hat{g}_\gamma]) \leq 4K \int_{\mathbb{R}^d} p_t(x)\, w_\gamma(x)^2\, \text{Tr}(\Sigma_t(x))\, dx + C_{\text{signal}}, \tag{6}$$

*where $C_{\text{signal}}$ represents the variance contribution from the learnable mean field.*

*Proof.* Let the per-sample loss for a target path connecting $x_0$ to $x_1$ be $J_t(\theta; x_1) = w_\gamma(x_t)\|v_\theta(x_t) - u_t(x_t|x_1)\|^2$. The stochastic gradient is $\hat{g}_\gamma = \nabla_\theta J_t = 2w_\gamma(x_t)(\nabla_\theta v_\theta)^\top(v_\theta - u_t(x_t|x_1))$. Using the Law of Total Variance, we decompose the variance over the marginal density $p_t(x)$:

$$\text{Var}[\hat{g}_\gamma] = \mathbb{E}_{x\sim p_t}[\text{Var}(\hat{g}_\gamma \mid x)] + \text{Var}_{x\sim p_t}[\mathbb{E}[\hat{g}_\gamma \mid x]].$$

We focus on the first term (intrinsic noise). For a fixed location $x$, the conditional variance is due to the variability of the target $u_t(x|x_1)$:

$$\text{Var}(\hat{g}_\gamma \mid x) = \mathbb{E}_{x_1|x}\left[\|\hat{g}_\gamma - \mathbb{E}[\hat{g}_\gamma|x]\|^2\right] \tag{7}$$

$$= 4w_\gamma(x)^2 \, \mathbb{E}_{x_1|x}\left[\left\|(\nabla_\theta v_\theta(x))^\top (u_t(x) - u_t(x|x_1))\right\|^2\right]. \tag{8}$$

Using the operator norm inequality $\|A^\top b\|^2 \leq \|A\|_{\text{op}}^2 \|b\|^2$, we have:

$$\text{Tr}(\text{Var}(\hat{g}_\gamma \mid x)) \leq 4w_\gamma(x)^2 \, \|\nabla_\theta v_\theta(x)\|_{\text{op}}^2 \, \text{Tr}(\Sigma_t(x)).$$

Applying the boundedness assumption $\|\nabla_\theta v_\theta(x)\|_{\text{op}}^2 \leq K$, we obtain:

$$\text{Tr}(\text{Var}(\hat{g}_\gamma \mid x)) \leq 4K w_\gamma(x)^2 \text{Tr}(\Sigma_t(x)).$$

Integrating this with respect to $p_t(x)$ yields the first term of the bound in Eq. (6). The second term ($C_{signal}$) corresponds to $\text{Var}_x[\mathbb{E}[\hat{g}_\gamma|x]]$, which depends on the learnable signal $u_t(x)$ and is independent of the conditional noise variance $\Sigma_t(x)$. □

In standard FM ($w_\gamma = 1$), the integral is dominated by the volume of the void space where $\Sigma_t(x)$ is large. By choosing $w_\gamma(x) \propto p_t(x)^\gamma$, the integrand becomes proportional to $p_t(x)^{1+2\gamma}\Sigma_t(x)$. Under the reasonable assumption that the signal ambiguity scales inversely with density (i.e., $\Sigma_t(x) \sim p_t(x)^{-\alpha}\Sigma_0$ for $\alpha > 0$), the $\gamma$-weighting with $\gamma \geq \alpha/2$ ensures that the noise contribution vanishes:

$$\lim_{p_t(x)\to 0} p_t(x)^{1+2\gamma}\Sigma_t(x) = 0.$$

This suggests that $\gamma$-FM suppresses gradient noise from the voids, concentrating the optimization budget on the high-density region.

**When is the variance–density scaling plausible?** The heuristic scaling $\Sigma_t(x) \propto p_t(x)^{-\alpha}$ is most plausible for conditional path designs in which the conditional target $u_t(x_t \mid x_1)$ becomes increasingly ill-conditioned in low-density regions of the marginal $p_t$. A representative example is Gaussian CFM, where $x_t$ is obtained by adding Gaussian noise and $u_t$ involves a score-like term of the intermediate marginal; in such settings the conditional variance of $u_t$ given $x_t$ typically increases as $p_t(x_t)$ decreases, reflecting amplification of estimation noise in "void" regions. More generally, for transport-noise interpolations that mix a deterministic drift toward $x_1$ with a stochastic perturbation, the signal-to-noise ratio of the conditional direction deteriorates away from the data manifold, so that $\text{tr}\,\Sigma_t(x)$ is larger where $p_t(x)$ is smaller. Our analysis in Proposition 3.1 should be read in this spirit: it formalizes how $\gamma$-weighting suppresses contributions from such high-variance, low-density regions, rather than requiring an exact power-law identity.

### 3.2 Physical Consistency with Liouville Dynamics

This subsection is intended as a geometric analogy, not as a derivation of the exact training dynamics of $\gamma$-FM. We use a classical model problem from optimal transport—the Wasserstein gradient flow of a Tsallis-type energy—to make precise a qualitative mechanism: density-power weighting suppresses motion in low-density ("void") regions. In particular, the resulting continuum dynamics exhibit a degenerate diffusion whose effective diffusivity vanishes as $p \to 0$, which leads to a finite-speed propagation effect.

Consider the generalized $\gamma$-entropy functional:

$$\mathcal{F}_\gamma[p] = \frac{1}{\gamma}\int p(x)^{\gamma+1}dx.$$

Notice that this functional corresponds exactly to the *self-divergence term* in the definition of the $\gamma$-divergence (up to a sign). Just as minimizing $\gamma$-divergence statistically ignores outliers as in the variance-reduction

argument above, the gradient flow of its associated entropy $\mathcal{F}_\gamma$ physically restricts the spread of probability mass.

The Wasserstein gradient flow is defined by the continuity equation driving the density along the gradient of the variation (see, e.g., Jordan et al. (1998); Ambrosio et al. (2008)).

$$\partial_t p = \nabla \cdot \left( p \nabla \frac{\delta \mathcal{F}_\gamma[p]}{\delta p} \right).$$

Calculating the variation $\frac{\delta \mathcal{F}_\gamma[p]}{\delta p} = \frac{\gamma+1}{\gamma} p^\gamma$ and substituting it gives the explicit dynamics:

$$\partial_t p(x,t) = \frac{\gamma+1}{\gamma} \nabla \cdot (p(x,t)^\gamma \nabla p(x,t)) = \Delta(p(x,t)^{1+\gamma}), \tag{9}$$

Equation (9) is the Wasserstein gradient flow of $\mathcal{F}_\gamma$ and will be used here as an exactly analyzable toy model that captures the qualitative effect of $\gamma$-weighting. For $\gamma > 0$, the diffusion is *degenerate*: the effective diffusivity scales like $p^\gamma$ and vanishes as $p \to 0$, which is the mechanism behind finite-speed propagation.

**Proposition 3.2** (Preservation of Compact Support). *Let $p(x,t)$ be the unique weak solution to the PME (9) with $\gamma > 0$, subject to a non-negative initial condition $p_0 \in L^1(\mathbb{R}^d) \cap L^\infty(\mathbb{R}^d)$. If the initial support $\mathrm{supp}(p_0)$ is compact, then the support $\mathrm{supp}(p(\cdot,t))$ remains compact for all $t > 0$. Moreover, there exists a constant $C$ depending on the initial mass such that the support is contained in a ball $\mathcal{B}(0, R(t))$ with*

$$R(t) \le C(1+t)^\beta, \quad \text{where } \beta = \frac{1}{d\gamma+2}.$$

*Proof.* The proof relies on the Comparison Principle for the Porous Medium Equation (Vázquez, 2007). The principle states that if two solutions $u$ and $v$ satisfy $u(x,0) \le v(x,0)$ everywhere, then $u(x,t) \le v(x,t)$ for all $t > 0$. We construct an explicit supersolution using the Barenblatt–Pattle solution $B(x,t;M)$, which represents the diffusion of a Dirac mass $M$. Let $B(x,\tau;M)$ be the Barenblatt profile centered at the origin with mass $M$ at a time shift $\tau > 0$:

$$B(x,\tau;M) = \tau^{-\alpha} \left( C(M) - \kappa \tau^{-2\beta}|x|^2 \right)_+^{1/\gamma},$$

where $(\cdot)_+ = \max(\cdot, 0)$. Since the initial data $p_0$ is bounded and has compact support, we can choose a sufficiently large mass $M$ and a time shift $\tau > 0$ such that the Barenblatt profile covers the initial data:

$$p_0(x) \le B(x,\tau) \quad \text{for all } x \in \mathbb{R}^d.$$

By the Comparison Principle, this ordering is preserved for all subsequent times $t > 0$:

$$p(x,t) \le B(x,t+\tau).$$

The support of the Barenblatt solution $B(\cdot, t+\tau)$ is explicitly known to be a ball of radius

$$R_B(t) = \sqrt{\frac{C(M)}{\kappa}}(t+\tau)^\beta.$$

Since $0 \le p(x,t) \le B(x,t+\tau)$, the support of $p$ must be contained within the support of $B$. Thus,

$$\mathrm{supp}(p(\cdot,t)) \subseteq \mathcal{B}(0, R_B(t)),$$

which implies that the support remains compact and expands at a rate of at most $O(t^\beta)$. In the limit $\gamma \to 0$, the exponent $\beta \to 1/2$, but the Barenblatt profile converges to a Gaussian which is strictly positive everywhere. Thus, the strict containment within a finite ball holds if and only if $\gamma > 0$. $\square$

In the toy model (9), the limit $\gamma \to 0$ reduces to the heat equation, whose solutions become instantly positive everywhere, reflecting infinite-speed propagation. By contrast, for $\gamma > 0$ the degeneracy at $p \approx 0$ yields an evolving interface and a finite-propagation behavior, as formalized in Proposition 3.2. While $\gamma$-FM does *not* literally solve (9), the same mechanism provides a useful intuition: when the weight behaves like $w_\gamma(x,t) \propto p_t(x)^\gamma$, updates are strongly down-weighted in low-density regions, and empirically this reduces spurious mass placed in "void" areas (a "void rejection" effect).

**Remark 3.3** (Generalized entropy and maximum entropy principle). *The functional*

$$\mathcal{F}_\gamma[p] = \frac{1}{\gamma} \int p(x)^{\gamma+1} \, dx$$

*is, up to an affine rescaling, the negative of the Tsallis q-entropy with $q = 1 + \gamma$ (Tsallis, 1988). It is well known that maximizing the Tsallis entropy under mass and second-moment constraints,*

$$\int p(x) \, dx = 1, \qquad \int \|x\|^2 p(x) \, dx = m_2,$$

*yields generalized Gaussian (or q-Gaussian) densities of the form*

$$p_q(x) = Z^{-1} \Big( 1 - (1-q)\beta \|x - \mu\|^2 \Big)_+^{\frac{1}{1-q}},$$

*for suitable parameters $\beta > 0$ and $\mu \in \mathbb{R}^d$. These q-Gaussians are precisely the self-similar Barenblatt profiles of the porous medium equation* (9) *for an appropriate correspondence between q and the nonlinearity exponent $1+\gamma$; see, e.g., Malacarne et al. (2001); Takatsu (2012). In particular, they exhibit compact support when $\gamma > 0$, providing concrete examples of the finite-support behaviour described in Proposition 3.2.*

### 3.3 Motivating Example: 1D Double-Well Potential

To visualize the macroscopic effect of our weighting scheme, it is instructive to consider a one-dimensional toy problem where the dynamics can be analyzed exactly. Consider a target distribution defined by a double-well potential $V(x) = \frac{1}{4}x^4 - \frac{3}{4}x^2$, where the target density satisfies $p_{data}(x) \propto \exp(-V(x))$. We analyze the probability flow transporting a standard Gaussian noise $\mathcal{N}(0,1)$ to this target.

The $\gamma$-weighted continuity equation can be mapped to a nonlinear Fokker–Planck equation with density-dependent diffusion:

$$\partial_t p_t = \nabla \cdot (p_t \nabla V) + D\nabla \cdot (p_t^\gamma \, \nabla p_t), \tag{10}$$

where $D$ is a diffusion constant. The behavior of this system critically depends on $\gamma$:

- **Case $\gamma = 0$ (Heat Equation):** The diffusion term becomes linear ($D\Delta p_t$). A fundamental property of the Heat Equation is its *infinite speed of propagation*. As illustrated in Figure 1 (Standard FM), probability mass instantaneously leaks into the high-potential barrier regions (voids) between the wells. This corresponds to the model "hallucinating" paths where no data exists.

- **Case $\gamma > 0$ (Porous Medium Equation):** Eq. (10) becomes the Porous Medium Equation (PME). A hallmark of the PME is its *finite speed of propagation*. The effective diffusivity $D_{\text{eff}} \propto p_t^\gamma$ vanishes in low-density regions. Consequently, the diffusion physically stops at the boundaries of the wells. As shown in Figure 1 ($\gamma$-FM), this creates a sharp geometric barrier that confines the flow to the main modes, preventing leakage into the void.

This analytical example should be read as a concrete illustration of the "Manifold Focusing" effect: our dynamic weighting $w_\gamma$ mimics the finite-propagation intuition of the PME and helps explain why the learned flow tends to avoid low-density voids, rather than as a derivation of the exact neural training dynamics.

### 3.4 Geometric Foundation: The $\gamma$-Stein Viewpoint

The physical behavior described by the PME (Section 3.2) is not accidental; it arises from the intrinsic geometry induced by the density weighting. While standard Flow Matching minimizes kinetic energy in a flat Euclidean geometry, we argue that $\gamma$-FM minimizes energy on a statistical manifold endowed with a density-dependent metric.

**The $\gamma$-Stein Metric.** From the perspective of Information Geometry (Eguchi, 2009), the $\gamma$-divergence generates a Riemannian metric structure on the space of probability densities. The $\gamma$-Stein operator for

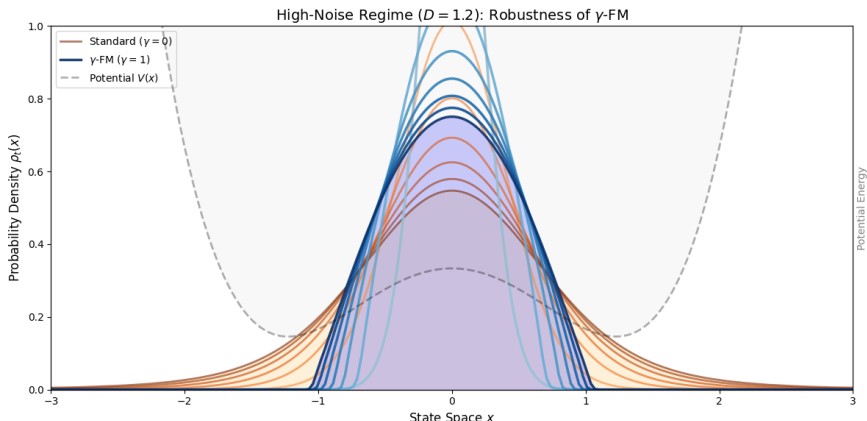

Figure 1: **Finite vs. Infinite Propagation.** Evolution of density under a double-well potential. **(Left) Standard FM ($\gamma = 0$):** Corresponds to linear diffusion (Heat Equation), causing probability mass to leak continuously into the low-density barrier (void). **(Right) $\gamma$-FM ($\gamma = 1$):** Corresponds to nonlinear diffusion (Porous Medium Equation) with finite propagation speed. The flow respects the potential barrier, keeping the mass tightly confined to the modes. This illustrates the mechanism of void rejection.

robust inference with unnormalized models was studied in Eguchi (2025). Following standard conventions in information geometry, we hereinafter denote the parametric family of model densities by $q_\theta$. The weighting $w_\gamma \approx q_\theta^\gamma$ in our objective (Eq. 2) naturally corresponds to the $\gamma$-*weighted Fisher information metric* $g^{(\gamma)}$ (Fujisawa & Eguchi, 2008; Matsuzoe, 2017):

$$g_{ij}^{(\gamma)}(\theta) = \int_{\mathbb{R}^d} \partial_{\theta_i} \log q_\theta(x)\, \partial_{\theta_j} \log q_\theta(x)\, q_\theta(x)^{1+\gamma} dx. \tag{11}$$

This metric measures distance based on the *escort measure* $d\mu_\gamma \propto q^{1+\gamma} dx$. Crucially, regions with low density $q(x) \approx 0$ make negligible contribution to the metric tensor. Geometrically, this means that "distances" in the void regions are compressed to zero, effectively removing them from the optimization landscape. The detailed discussion is given in Appendix A.

**Flow Matching as Geodesic Optimization.** Let $\{q_\theta : \theta \in \Theta\}$ be a parametric family of densities on $\mathbb{R}^d$, and let $t \mapsto \theta(t)$ be a time-dependent curve in parameter space with associated path of densities $q_{\theta(t)}$. A probability flow $(q_{\theta(t)}, v_t)_{t \in [0,1]}$ satisfies the continuity equation

$$\partial_t q_{\theta(t)}(x) + \nabla \cdot \big(q_{\theta(t)}(x)\, v_t(x)\big) = 0.$$

For a fixed $\theta$, we equip the space of vector fields with the $\gamma$-weighted inner product

$$\langle u, v \rangle_{\theta, \gamma} := \int_{\mathbb{R}^d} u(x)^\top v(x)\, q_\theta(x)^{1+\gamma}\, \mathrm{d}x,$$

and denote by $L_\gamma^2(q_\theta)$ the corresponding Hilbert space. The $\gamma$-Stein operator associated with $q_\theta$ is defined by

$$\mathcal{A}_{q_\theta}^{(\gamma)} f(x) := q_\theta(x)^{-1}\, \nabla \cdot \big(q_\theta(x)^{\gamma+1} f(x)\big), \qquad f : \mathbb{R}^d \to \mathbb{R}^d.$$

We regard the tangent space of the statistical manifold at $\theta$ as the closure (in $L_\gamma^2(q_\theta)$) of the range of $\mathcal{A}_{q_\theta}^{(\gamma)}$:

$$T_\theta \mathcal{M}_\gamma := \overline{\big\{ \mathcal{A}_{q_\theta}^{(\gamma)} f : f \in C_c^\infty(\mathbb{R}^d, \mathbb{R}^d) \big\}}.$$

In other words, $T_\theta \mathcal{M}_\gamma$ consists of all $\gamma$-Stein-type velocity fields that preserve total mass under the weighted geometry.

The $\gamma$-weighted flow-matching objective can be written as

$$\mathcal{L}_\gamma = \int_0^1 \left\| v_t - v_t^* \right\|_{L^2_\gamma(q_{\theta(t)})}^2 \, \mathrm{d}t,$$

where $v_t^*$ is the ideal velocity field induced by the target transport. For each $t$, the minimizer $u_t$ of $\mathcal{L}_\gamma$ over $T_{\theta(t)}\mathcal{M}_\gamma$ is the orthogonal projection of $v_t^*$ onto the tangent space with respect to $\langle \cdot, \cdot \rangle_{\theta(t),\gamma}$. This motivates the definition of the projection operator

$$\Pi_{\theta,\gamma} : L^2_\gamma(q_\theta) \to T_\theta \mathcal{M}_\gamma, \qquad \Pi_{\theta,\gamma}[f] := \arg \min_{u \in T_\theta \mathcal{M}_\gamma} \left\| f - u \right\|_{L^2_\gamma(q_\theta)}^2.$$

In particular, $u_t = \Pi_{\theta(t),\gamma}[v_t^*]$ is the $\gamma$-Stein projection of the ideal velocity field onto the statistical manifold.

Following Appendix A, the $\gamma$-Stein connection $\nabla^{(\gamma)}$ is characterised by the requirement that the covariant derivative of a time-dependent velocity field $u_t$ along a curve $t \mapsto \theta(t)$ is obtained by projecting the ordinary time derivative back onto the tangent space:

$$D_t^{(\gamma)} u_t := \Pi_{\theta(t),\gamma}\big[\partial_t u_t\big].$$

A curve $\theta(t)$ is a (parametric) geodesic of the $\gamma$-Stein geometry if its associated velocity field $u_t \in T_{\theta(t)}\mathcal{M}_\gamma$ is covariantly constant, that is,

$$D_t^{(\gamma)} u_t = \Pi_{\theta(t),\gamma}\big[\partial_t u_t\big] = 0 \quad \text{for all } t \in [0,1].$$

In this idealized geometry, minimizing $\mathcal{L}_\gamma$ over admissible flows $\big(q_{\theta(t)}, u_t\big)_{t \in [0,1]}$ may be interpreted as searching for geodesic-like curves on the statistical manifold endowed with the $\gamma$-Stein metric and connection. The case $\gamma = 0$ reduces to the flat $L^2$ geometry, where the tangent space coincides with $L^2(q_{\theta(t)})$ and $\Pi_{\theta(t),0}$ is the identity, so that geodesics correspond to straight lines and the velocity field must be defined everywhere. For $\gamma > 0$, the metric is weighted by $q_{\theta(t)}^{1+\gamma}$, so that directions supported in low-density regions have negligible norm. This does not prove that the implemented algorithm exactly follows geodesics of the escort geometry; rather, it motivates why the learned velocity field tends to concentrate on the high-density manifold through the $\gamma$-Stein projection.

### 3.5 Implicit Geometric Regularization

We now formalize the implicit-regularization intuition stated earlier by deriving a Sobolev-type roughness functional induced by the $\gamma$-weighted objective. Let $J_\theta(x,t) = \nabla_x v_\theta(x,t)$ be the Jacobian of the model. The stiffness of the ODE solver is controlled by the Lipschitz constant $L(t) \approx \sup_x \|J_\theta(x,t)\|_F$.

In unweighted regression, minimizing the loss in voids (where the target $u_t$ is chaotic) requires $v_\theta(x,t)$ to change rapidly, driving $\|J_\theta(x,t)\|_F$ to be large. We formalize the regularization effect as a bound on the weighted Sobolev norm:

$$\mathcal{R}_{\text{roughness}}(t) := \int_{\mathbb{R}^d} q_{\theta(t)}(x) w_\gamma(x,t) \|\nabla_x v_\theta(x,t)\|_F^2 \, dx. \tag{12}$$

By downweighting the voids ($w_\gamma(x,t) \to 0$), $\gamma$-FM relaxes the constraint on $v_\theta(x,t)$ in these regions. Assuming the neural network has a spectral bias towards low-frequency functions, removing the high-frequency targets in the voids implies that the minimizer $v_\theta^*(x,t)$ will effectively default to a smooth interpolation in the ambient space.

Empirically, we observe a significant reduction in the Jacobian norm within the ambient space:

$$\mathbb{E}_{x \sim p_{\text{ambient}}} \left[ \|\nabla_x v_\theta^{(\gamma>0)}(x,t)\|_F \right] \ll \mathbb{E}_{x \sim p_{\text{ambient}}} \left[ \|\nabla_x v_\theta^{(0)}(x,t)\|_F \right],$$

where $p_{\text{ambient}}$ represents the distribution of the void regions (e.g., uniform noise in the bounding box). This reduction in the local Lipschitz constant implies a less stiff ODE, permitting adaptive solvers to take larger integration steps. This mechanism is consistent with the improved NFE behavior observed in our experiments, although we do not claim a universal causal guarantee beyond the tested setting.

**A Dirichlet–spectral perspective.** To connect the empirical reduction in (12) with a more geometric picture, it is convenient to introduce the weighted Dirichlet form associated with the escort weight. Such variational limits of discrete regularizers on data manifolds have been rigorously studied in the context of Optimal Transport by Hamm et al. (2025). Fix a time $t$ and write $p_t$ for the marginal density of $x_t$. Using the weighting factor $w_\gamma(x, t) \propto q_{\theta(t)}(x)^\gamma$, we introduce the escort measure

$$\mathrm{d}\mu_{\gamma,t}(x) := q_{\theta(t)}(x) w_\gamma(x, t) \, \mathrm{d}x.$$

We then define the weighted Dirichlet form associated with this measure:

$$\mathcal{E}_{\gamma,t}(f, f) := \int_{\mathbb{R}^d} \|\nabla_x f(x)\|^2 \, \mathrm{d}\mu_{\gamma,t}(x). \tag{13}$$

By integration by parts, this form is associated with the self-adjoint operator

$$\mathcal{L}_{\gamma,t} f := q_{\theta(t)}^{-1} \nabla_x \cdot \big(q_\gamma(x, t) \, \nabla_x f(x)\big), \tag{14}$$

with $q_\gamma(x, t) = q_{\theta(t)}(x) w_\gamma(x, t)$ in the weighted Hilbert space $L^2(\mu_{\gamma,t})$. The Poincaré inequality for $\mu_{\gamma,t}$ can then be written as

$$\int_{\mathbb{R}^d} \big(f(x) - \bar{f}_{\gamma,t}\big)^2 \, \mathrm{d}\mu_{\gamma,t}(x) \leq \frac{1}{\lambda_{\gamma,t}} \, \mathcal{E}_{\gamma,t}(f, f), \qquad \bar{f}_{\gamma,t} := \int f \, \mathrm{d}\mu_{\gamma,t}, \tag{15}$$

where $\lambda_{\gamma,t} > 0$ is the spectral gap, that is, the smallest positive eigenvalue of $-\mathcal{L}_{\gamma,t}$. When $q_{\theta(t)}$ is strongly log-concave with curvature lower bound $\kappa > 0$, the escort measure $\mu_{\gamma,t}$ inherits a stronger curvature of order $(1 + \gamma)\kappa$, and the spectral gap $\lambda_{\gamma,t}$ grows at least linearly in $(1 + \gamma)$.

To make the link with (12) more explicit, let us consider an idealized, regularized regression problem at a fixed time $t$ and for a single scalar coordinate of the vector field. Let $u_t$ denote the target component and consider functions $f : \mathbb{R}^d \to \mathbb{R}$. We introduce the Tikhonov-regularized objective

$$\mathcal{J}_{\gamma,t}(f) := \int_{\mathbb{R}^d} q_{\theta(t)}(x) w_\gamma(x, t) \big(f(x) - u_t(x)\big)^2 \, \mathrm{d}x + \tau \, \mathcal{E}_{\gamma,t}(f, f), \qquad \tau > 0, \tag{16}$$

which is the population analogue of a $\gamma$-FM regression loss with an explicit weighted Sobolev penalty. The next proposition studies an idealized population objective where we add an explicit weighted Sobolev penalty to the $\gamma$-FM loss. Although Algorithm 1 does not explicitly add a Tikhonov penalty, the analysis above identifies a roughness functional naturally associated with the $\gamma$-weighted objective. In practice, this provides a mechanistic explanation for the observed smoothness improvements, which may be further amplified by the implicit bias of SGD (see, e.g., Rahaman et al. (2019); Xu et al. (2020); Jin & Montúfar (2023)). The Dirichlet-regularized objective in (16) should therefore be viewed as a tractable surrogate that makes the geometric effect of the escort weighting explicit in the eigenbasis of the weighted Laplacian. Let $f^*_{\gamma,t}$ denote the unique minimizer of $\mathcal{J}_{\gamma,t}$.

**Proposition 3.4** (Spectral shrinkage under $\gamma$-weighted Dirichlet regularization). *Let $\{\varphi_k^{(\gamma,t)}\}_{k \geq 0}$ be an orthonormal eigenbasis of $L^2(\mu_{\gamma,t})$ consisting of eigenfunctions of $-\mathcal{L}_{\gamma,t}$,*

$$-\mathcal{L}_{\gamma,t} \, \varphi_k^{(\gamma,t)} = \mu_k^{(\gamma,t)} \, \varphi_k^{(\gamma,t)}, \qquad 0 = \mu_0^{(\gamma,t)} < \mu_1^{(\gamma,t)} \leq \mu_2^{(\gamma,t)} \leq \cdots. \tag{17}$$

*Expand the target as*

$$u_t(x) = \sum_{k \geq 0} b_k^{(\gamma,t)} \, \varphi_k^{(\gamma,t)}(x). \tag{18}$$

*Then the minimizer $f^*_{\gamma,t}$ of (16) admits the expansion*

$$f^*_{\gamma,t}(x) = \sum_{k \geq 0} a_k^{(\gamma,t)} \, \varphi_k^{(\gamma,t)}(x), \qquad a_k^{(\gamma,t)} = \frac{1}{1 + \tau \, \mu_k^{(\gamma,t)}} \, b_k^{(\gamma,t)}. \tag{19}$$

*Moreover, its Dirichlet energy is given by*

$$\mathcal{E}_{\gamma,t}\big(f^*_{\gamma,t}, f^*_{\gamma,t}\big) = \sum_{k \geq 1} \frac{\mu_k^{(\gamma,t)}}{\big(1 + \tau \, \mu_k^{(\gamma,t)}\big)^2} \, \big(b_k^{(\gamma,t)}\big)^2. \tag{20}$$

*Proof.* Since (16) and (13) are quadratic and diagonalizable in the eigenbasis $\{\varphi_k^{(\gamma,t)}\}$, we write

$$f(x) = \sum_{k\geq 0} a_k\,\varphi_k^{(\gamma,t)}(x), \qquad u_t(x) = \sum_{k\geq 0} b_k\,\varphi_k^{(\gamma,t)}(x).$$

Orthogonality in $L^2(\mu_{\gamma,t})$ gives

$$\int q_{\theta(t)}(x) w_\gamma(x,t)\left(f(x) - u_t(x)\right)^2 \mathrm{d}x = \sum_{k\geq 0}\left(a_k - b_k\right)^2,$$

while (13) and (17) imply

$$\mathcal{E}_{\gamma,t}(f,f) = \sum_{k\geq 1} \mu_k^{(\gamma,t)}\, a_k^2.$$

Therefore

$$\mathcal{J}_{\gamma,t}(f) = \sum_{k\geq 0}\left[\left(a_k - b_k\right)^2 + \tau\,\mu_k^{(\gamma,t)}\, a_k^2\right]$$

splits into independent one-dimensional problems in the coefficients $a_k$. Minimizing each term over $a_k$ yields

$$2\left(a_k - b_k\right) + 2\tau\,\mu_k^{(\gamma,t)}\, a_k = 0, \qquad \Rightarrow \quad a_k = \frac{1}{1 + \tau\,\mu_k^{(\gamma,t)}}\, b_k,$$

which gives (19). Substituting back into $\mathcal{E}_{\gamma,t}(f,f)$ directly yields (20). $\qquad\square$

In the case of $\gamma = 0$, the measure $\mu_{0,t}$ reduces to the standard density $q_{\theta(t)}$, and Eq. (19) recovers the standard spectral filtering result known in manifold regularization (Belkin et al., 2005). The significance of Proposition 3.4 lies in the dependence on $\gamma$: as discussed in the proof, increasing $\gamma$ effectively rescales the eigenvalues $\mu_k^{(\gamma,t)}$, thereby intensifying the shrinkage effect on high-frequency modes compared to the standard case. We suggest a close relation to the Witten Laplacian and the Bakry–Émery curvature in the proof. Assume that the marginal density $q_{\theta(t)}$ admits a smooth potential $U_t$ such that $q_{\theta(t)}(x) = \exp(-U_t(x))$. Then the generator $L_{\gamma,t}$ in (14) can be rewritten as

$$L_{\gamma,t} f(x) = \Delta f(x) + \langle \nabla_x \log w_\gamma(x,t), \nabla_x f(x)\rangle = \Delta f(x) - (1 + \gamma)\,\langle \nabla_x U_t(x), \nabla_x f(x)\rangle,$$

which is the Witten (or Bakry–Émery) Laplacian associated with the potential $(1+\gamma)U_t$. In the Bakry–Émery $\Gamma_2$ calculus, the curvature-dimension condition

$$\nabla_x^2 U_t(x) \succeq \kappa I_d \qquad (\kappa > 0)$$

implies that the carré du champ $\Gamma(f) = \|\nabla_x f\|^2$ and its iterated form $\Gamma_2(f)$ satisfy

$$\Gamma_2(f) \;:=\; \frac{1}{2}\Big(L_{\gamma,t}\Gamma(f) - 2\langle \nabla_x f, \nabla_x L_{\gamma,t} f\rangle\Big) \;\geq\; (1+\gamma)\kappa\,\Gamma(f) \quad \forall f.$$

As a consequence, the measure $\mu_{\gamma,t}$ satisfies the Poincaré inequality (15) with a spectral gap bounded below by

$$\lambda_{\gamma,t} \;\geq\; (1+\gamma)\kappa.$$

Thus the escort reweighting $w_\gamma \propto q_{\theta(t)}^{1+\gamma}$ simply rescales the potential in the Witten Laplacian, amplifying curvature and enlarging the spectral gap. This provides a geometric justification for our heuristic that the eigenvalues $\mu_k^{(\gamma,t)}$ of $-L_{\gamma,t}$ grow approximately linearly in $(1 + \gamma)$, and therefore the high-frequency modes in the Dirichlet energy (20) are increasingly damped as $\gamma$ increases.

It is noted that the factor

$$F(\mu) := \frac{\mu}{\left(1 + \tau\,\mu\right)^2}$$

governs the contribution of an eigenmode with eigenvalue $\mu$ to the roughness (20). A direct calculation shows

$$F'(\mu) = \frac{1 - \tau\,\mu}{\left(1 + \tau\,\mu\right)^3},$$

so that $F'(\mu) < 0$ whenever $\mu > 1/\tau$. In other words, for sufficiently high-frequency modes (large eigenvalues $\mu_k^{(\gamma,t)}$) the Dirichlet contribution

$$\frac{\mu_k^{(\gamma,t)}}{\left(1 + \tau\,\mu_k^{(\gamma,t)}\right)^2}\left(b_k^{(\gamma,t)}\right)^2$$

is a decreasing function of $\mu_k^{(\gamma,t)}$. Under curvature assumptions on $q_{\theta(t)}$, the escort operator $-\mathcal{L}_{\gamma,t}$ has eigenvalues that increase with $\gamma$ (roughly $\mu_k^{(\gamma,t)} \approx (1+\gamma)\,\mu_k^{(0,t)}$), so that high-frequency contributions to (20) are systematically damped as $\gamma$ grows. If the target field $u_t$ carries most of its energy in such high-frequency modes, the total roughness $\mathcal{E}_{\gamma,t}(f_{\gamma,t}^*, f_{\gamma,t}^*)$ decreases as a function of $\gamma$.

In practice, the neural network vector field $v_\theta(x,t)$ is not explicitly regularized by (16). However, stochastic gradient descent with a finite-capacity network is known to exhibit an implicit bias towards functions with small Sobolev norm. The above spectral calculation suggests that the $\gamma$-dependent escort geometry further amplifies this bias on the data manifold: the weighted roughness

$$\mathcal{R}_{\text{roughness}} = \int_{\mathbb{R}^d} q_{\theta(t)}(x) w_\gamma(x,t)\,\|\nabla_x v_\theta(x,t)\|_F^2\,\mathrm{d}x \tag{21}$$

is dominated by high-frequency modes whose eigenvalues increase with $\gamma$, and these modes are exactly those that are most strongly damped by the effective Tikhonov term. This provides a theoretical explanation for the empirical trend observed in the following section, where the smoothness metric decreases as $\gamma$ increases, and supports the interpretation of $\gamma$-FM as a geometrically informed regularizer that suppresses oscillatory behavior in void regions while preserving expressiveness near the data manifold. In practice we do not add the Dirichlet term explicitly; instead the combination of spectral bias and finite training time acts as an effective Sobolev regularizer.

### 3.6 Theoretical Selection of $\gamma$ via Geometric Selection Criterion

A critical practical question is the selection of the density-weighting parameter $\gamma$. Typically, the method of cross-validation is employed, but the computation for the current task is expensive and infeasible. Accordingly, we construct a tractable *function-space proxy* suitable for Flow Matching, which we term the Geometric Selection Criterion (GSC):

$$\text{GSC}(\gamma) = \text{MMD}^2(p_{\text{data}}, p_{\theta_\gamma}) + \lambda \mathcal{R}_{\text{roughness}}(v_{\theta_\gamma}), \tag{22}$$

where $\lambda > 0$ is a trade-off parameter (we set $\lambda = 1$). Here, the squared Maximum Mean Discrepancy (MMD) serves as the bias proxy, measuring generation quality. The roughness functional $\mathcal{R}_{\text{roughness}}(v_{\theta_\gamma})$ derived in (21) serves as the stability penalty. Minimizing the GSC may help identify a $\gamma$ that balances faithful data reconstruction and geometric regularity.

## 4 Experiments

### 4.1 Synthetic Verification: Implicit Regularization

Before evaluating our method on complex image datasets, we first verify the "Implicit Geometric Regularization" hypothesis (Section 3.5) in a controlled high-dimensional setting. A fundamental challenge in Flow Matching is the "curse of dimensionality": as the dimension $D$ increases, the relative volume of the data manifold vanishes, and the vast majority of the integration domain becomes empty "void" space.

**Experimental Setup.** To make the weighting mechanism visually transparent, we construct a ring-shaped target distribution in a high-dimensional ambient space $\mathbb{R}^{20}$. The first two coordinates contain the ring

Figure 2: Where the shared-time weighting proxy acts in the ring toy experiment. Panel (a) shows a two-dimensional slice of the interpolation-path density at a late shared time $t = 0.85$, where the path mass is concentrated near the ring support. Panel (b) shows the corresponding practical rank-based weighting proxy $w_\gamma(x_t)$, which places larger emphasis near the ring-support region and downweights both the center void and the outside-support region. Panels (c) and (d) compare the learned speed maps of standard FM and $\gamma$-FM, respectively. Panel (e) shows radial profiles of the path density, the weighting proxy, and the learned speeds. The figure is intended to separate where the path places mass, where the weighting proxy acts, and how the learned field changes as a result.

structure, while the remaining coordinates consist of low-level Gaussian noise. We then train both standard FM ($\gamma = 0$) and $\gamma$-FM ($\gamma = 1$), and visualize a two-dimensional slice of the learned fields. For this toy experiment, we use the same practical shared-time ordering surrogate as in the main experiments, and inspect a late interpolation slice at $t = 0.85$ so that the ring-support region, the center void, and the outside-support region are visually separated.

**Results: Where the weight acts.** Figure 2 is designed to separate three distinct objects: the location of path mass, the location where the practical weighting proxy acts, and the geometry of the resulting learned field. Panel (a) shows that at the displayed late time the interpolation-path density is concentrated near the ring. Panel (b) shows the corresponding rank-based monotone weighting proxy, which places larger emphasis near that support region and downweights both the center void and the far exterior. Panels (c) and (d) then compare the learned speed maps under FM and $\gamma$-FM, while panel (e) summarizes the same comparison through radial profiles.

The main qualitative message is not that the practical surrogate is a literal density estimator, but that it provides a useful within-batch ordering signal. In this toy example, that ordering concentrates learning pressure near the ring-support region and weakens it in low-density regions. As a result, the learned field under $\gamma$-FM is visually smoother in the center void and the outside-support region than the field learned by standard FM. This revised visualization therefore clarifies where the weighting acts and how it changes the learned geometry, which was less transparent in the previous version of the figure.

## 4.2 Latent-flow modelling of CIFAR-10

Our experimental setting follows the general latent-flow paradigm of Lipman et al. (2023) in the sense that we train a flow in the latent space of a pre-trained autoencoder rather than directly in pixel space. Concretely, we first train an autoencoder on CIFAR-10 and then freeze the encoder and decoder; all flow-matching experiments are carried out in the resulting latent space.

In contrast to Lipman et al. (2023), who employ the standard (unweighted) flow-matching objective and focus on high-resolution image synthesis, we keep the latent architecture fixed and vary only the *regression geometry* in latent space via the $\gamma$-weighted loss (2). Thus, the comparisons in Table 1 and the accompanying discussion are designed so that differences in MMD, smoothness, NFE, and decoded-image metrics can be attributed to the effect of the $\gamma$-weighting rather than to changes in the autoencoder or flow architecture.

## 4.3 Evaluation on latent CIFAR-10

To address whether the observed gains of $\gamma$-FM arise specifically from density-weighted geometry rather than from generic smoothing alone, we expand the latent CIFAR-10 evaluation in four directions. Throughout this subsection, we work under a *robust-target* latent protocol rather than a purely clean one: the flow model is trained on an inlier–outlier mixture in latent space, while decoded-image metrics are evaluated against a held-out inlier-class test set. This setting allows us to study whether density weighting organizes the regression geometry toward the dominant data manifold, instead of merely acting as a generic smoother. As a complementary sanity check, we also repeated the same latent CIFAR-10 pipeline under a clean-target protocol, in which the flow model is trained on inlier latents only. That rerun is useful for separating two questions: whether the practical weighting mechanism remains operational on clean data, and whether it yields a measurable gain when the regression geometry is not distorted by contaminated or off-support targets. Empirically, the weighting remained active in the clean rerun, but the performance gain largely disappeared, with shared-time FM already competitive in FID, Recall, and Coverage. We therefore use the robust-target setting as the main empirical regime for evaluating the practical advantage of the proposed method.

First, we build the training protocol so that the dynamic density proxy is computed from a *shared-time minibatch*, as described in Section 2.3 and Algorithm 1. This ensures that the $k$-NN statistic is computed within a particle cloud sampled from a single marginal $p_t$ induced by the robust-target training distribution.

Second, we add an explicit regularization baseline:

$$\text{FM} + \text{Jac},$$

which augments the standard flow-matching loss with a Jacobian penalty,

$$\mathcal{L}_{\text{FM+Jac}}(\theta) = \mathcal{L}_{\text{FM}}(\theta) + \lambda_J \, \mathbb{E}_{t,x,\varepsilon} \big[ \| J_x v_\theta(x,t)^\top \varepsilon \|^2 \big],$$

where $\varepsilon \sim \mathcal{N}(0, I)$ (or a Rademacher probe) and the Jacobian norm is estimated by a Hutchinson-type trace estimator. This baseline is trained with the same architecture, optimizer, number of updates, and ODE solver settings as the proposed method.

Third, beyond latent-space discrepancy and vector-field smoothness, we report image-space generative metrics after decoding. Specifically, we evaluate:

- **FID** (lower is better), to assess overall sample quality with respect to the inlier-class reference set;

- **Recall** (higher is better), to quantify diversity;

- **Coverage** (higher is better), to assess mode support.

These metrics complement the latent-space RBF-MMD and the ambient roughness proxy

$$\tilde{R}_{\text{roughness}}(v_\theta) = \mathbb{E}_{x \sim \mathcal{N}(0,I)} \big[ \| \nabla_x v_\theta(x) \|_F^2 \big],$$

thereby allowing us to examine the trade-off between vector-field smoothness, robustness to off-manifold targets, and inlier-oriented decoded-image quality.

Fourth, we make the model-selection protocol explicit. We compare the following models under matched computational budgets:

$$\text{FM}, \qquad \gamma\text{-FM (shared-}t), \qquad \text{FM} + \text{Jac},$$

and optionally also report $\gamma$-FM+Jac when discussing whether density-weighting and explicit regularization are complementary. For fairness, the primary comparison is made under the same network, optimizer, training steps, and the same fixed-step midpoint solver budget. We select $\gamma$ from a fixed grid and $\lambda_J$ from a matched regularization grid using validation FID, and when possible we also report a *matched-smoothness* comparison to isolate the smoothness/diversity trade-off.

Table 1: Shared-time CIFAR-10 comparison under the robust-target latent protocol and matched computational budgets. All methods use the same latent autoencoder, the same flow backbone, the same number of training updates, and the same ODE solver budget.

| Method | FID↓ | Recall↑ | Coverage↑ | $\text{MMD}^2_{\text{in}}$ ↓ | $\text{MMD}^2_{\text{out}}$ ↓ | Smoothness↓ |
|---|---|---|---|---|---|---|
| FM (shared-$t$) | 59.063 | 0.388 | 0.915 | 0.000609 | 0.012659 | 9.136 |
| $\gamma$-FM ($\gamma = 0.2$) | 59.761 | 0.390 | 0.911 | 0.000631 | 0.012531 | 9.156 |
| $\gamma$-FM ($\gamma = 0.5$) | 60.609 | 0.382 | 0.916 | 0.000685 | 0.012046 | 9.218 |
| $\gamma$-FM ($\gamma = 1.0$) | 61.198 | 0.395 | 0.915 | 0.000773 | 0.011684 | 8.788 |
| $\gamma$-FM ($\gamma = 2.0$) | 62.178 | 0.387 | 0.908 | 0.001060 | 0.011118 | 7.046 |
| FM+Jac | 60.305 | 0.397 | 0.924 | 0.000598 | 0.012915 | 9.709 |

**Observed trade-off under the shared-time surrogate.** Table 1 shows the trade-off induced by density weighting under the shared-time protocol in the robust-target latent setting. As $\gamma$ increases, the out-of-support discrepancy decreases monotonically, and the smoothness proxy improves overall, indicating progressively stronger suppression of unstable low-density updates. At the same time, FID with respect to the inlier-class reference set tends to worsen and $\text{MMD}^2_{\text{in}}$ tends to increase, showing that stronger weighting is not free in terms of inlier-oriented fidelity. In the present robust-target latent CIFAR-10 study, $\gamma = 0.5$ provides a reasonable compromise between low-density suppression and decoded-image quality.

A complementary clean-target rerun helps interpret this result. In that setting, the weighting mechanism remained active, but the empirical advantage largely disappeared: shared-time FM was already competitive, and $\gamma$-FM became mostly comparable or slightly inferior in FID, Recall, and Coverage, while still tending to produce somewhat smoother vector fields. We therefore interpret the main gain of $\gamma$-weighting not as a universal improvement on all latent-generation tasks, but as a geometry-dependent benefit that becomes most visible when contaminated or low-support targets distort the regression problem.

The FM+Jac baseline is informative because it represents explicit smoothing under the same training budget. Empirically, FM+Jac improves Recall and Coverage, but it does not reproduce the same reduction in $\text{MMD}^2_{\text{out}}$ achieved by larger values of $\gamma$. This supports our interpretation that $\gamma$-weighting is not merely a generic smoothness penalty, but a change in regression geometry that selectively deemphasizes low-density regions.

**Comparison with explicit Jacobian regularization.** The FM+Jac baseline is informative but exhibits a different profile from $\gamma$-FM. In the shared-time runs, explicit Jacobian penalization could improve in-support fit and in some cases slightly helped Recall or Coverage, but it did not reproduce the systematic reduction in out-of-support discrepancy obtained by density weighting. Its effect on smoothness and FID was also less consistent, and the seed-to-seed variability was generally larger. This suggests that Jacobian regularization acts mainly as a generic local smoother, whereas $\gamma$-weighting changes the regression geometry by selectively suppressing low-density updates. We therefore regard FM+Jac as a useful comparison baseline, but not as a replacement for the void-rejection mechanism induced by the density-weighted objective.

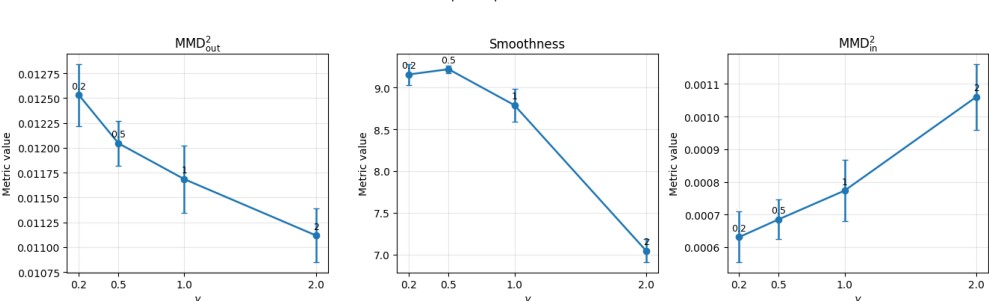

Figure 3: Shared-time $\gamma$-sweep on latent CIFAR-10 under the robust-target protocol and the rank-based piecewise surrogate. Left: $\mathrm{MMD}^2_{\mathrm{out}}$ decreases as $\gamma$ increases, indicating improved suppression of out-of-support trajectories. Middle: the smoothness proxy improves overall as $\gamma$ becomes larger, consistent with a more conservative vector field in low-density regions. Right: $\mathrm{MMD}^2_{\mathrm{in}}$ increases with $\gamma$, showing the accompanying cost in in-support fidelity. Together, these panels visualize the trade-off reported in Table 1. A separate clean-target rerun showed that the weighting mechanism remained active there as well, but the gain in decoded-image quality largely disappeared, so the present robust-target regime is the one in which the advantage of density weighting is most clearly expressed.

**Adaptive selection via GSC and validation metrics.** To choose $\gamma$ without relying only on visual inspection, we continue to examine the Geometric Selection Criterion (GSC) from Section 3.6, but we interpret it as a practical heuristic rather than as a definitive optimality principle. In practice, we read the GSC curve together with validation FID and decoded-image diagnostics, so that the choice of $\gamma$ is supported by both latent-space and decoded-image summaries. In the present robust-target CIFAR-10 latent study, this combined view suggests $\gamma = 0.5$ as a reasonable operating point, while $\gamma = 1.0$ and $2.0$ are most useful for illustrating increasingly conservative low-density behavior.

**Smoothness–diversity trade-off under the shared-time surrogate.** Figure 3 complements Table 1 by visualizing the same trade-off more directly. As $\gamma$ increases, the model becomes more conservative in low-density regions under the robust-target protocol, which improves the out-of-support discrepancy and the smoothness proxy, while increasing the in-support discrepancy. In this sense, the shared-time protocol reveals a coherent trade-off rather than a uniformly dominant improvement. Among the tested values, $\gamma = 0.5$ provides a reasonable operating point in our robust-target latent CIFAR-10 experiments, whereas $\gamma = 1.0$ and especially $\gamma = 2.0$ make the same tendency more pronounced.

Viewed together with the clean-target sanity check, this figure also clarifies the regime in which the weighting is most useful. When the training targets are already clean, the same weighting mechanism still operates, but its effect is expressed mainly as additional regularization rather than as a clear gain in decoded-image quality. This supports the interpretation that the proposed method is particularly useful when the regression geometry is perturbed by contamination or by low-support target regions.

Thus, the GSC may provide a useful geometric diagnostic for increasingly conservative low-density regularization, even though in the present experiments it is not used as a stand-alone selection rule. Future work may explore scalable approximations of the rigorous penalty, such as diagonal approximations or last-layer Laplace approximations, to bridge the gap between the rigorous theory and deep learning practice.

**Practical stability of the shared-time proxy.** A natural concern is whether the shared-time $k$-NN weighting proxy becomes unreliable when the minibatch is small. To address this point directly, we report an additional minibatch-size ablation in Appendix D.1, where the latent CIFAR-10 experiment is repeated under the same frozen autoencoder, decoding pipeline, optimizer, and training budget while varying only the minibatch size. The main finding is that, in the tested latent-CIFAR regime, the proxy retains a stable nontrivial dynamic range across the tested batch sizes rather than collapsing at smaller minibatches. At the same time, both FM and $\gamma$-FM improve as the batch size increases, indicating that the degradation at small

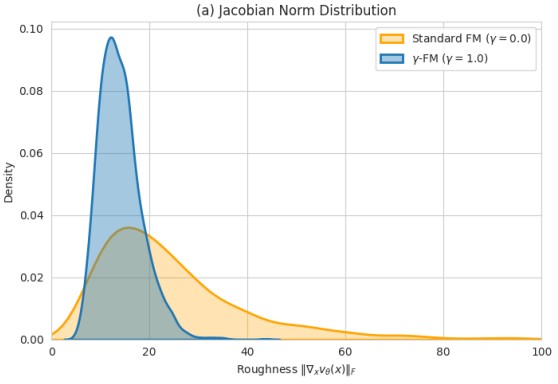

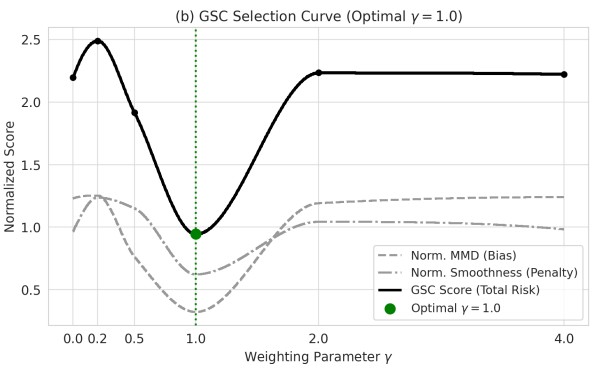

(b) GSC selection curve

(a) Jacobian norm distribution

Figure 4: Auxiliary diagnostics for smoothness and $\gamma$ selection. (a) **Micro-level analysis:** Illustrative roughness histogram for the baseline ($\gamma = 0$) and our method ($\gamma = 1.0$). The weighted objective is associated with a milder roughness profile and fewer extreme oscillatory regions, although the two distributions still overlap substantially. (b) **Macro-level selection:** The GSC score across different $\gamma$ values. In the shared-time protocol, this curve is interpreted jointly with validation FID and decoded-image diagnostics rather than as a stand-alone optimum certificate.

$B$ is not specific to the density-weighted objective. In particular, Appendix D.1 shows that the final ESS$/B$ stays near 0.52 across the tested batch sizes, indicating that the proxy retains a stable nontrivial dynamic range rather than degenerating into nearly uniform or highly singular weights. We therefore interpret the shared-time surrogate as a practically stable within-batch ordering device in the regime studied here, rather than as a general guarantee for high-dimensional minibatch density estimation. Moreover, the same ablation shows that at low to moderate batch sizes, $\gamma$-FM tends to produce smoother learned vector fields than FM while maintaining broadly comparable recall and coverage.

### 4.4 Stress test under contaminated latents

While the density-weighted objective is well defined under both clean and contaminated targets, the clean-target rerun suggests that its empirical benefit is most visible when the regression geometry is distorted by out-of-support or contaminated latents. We therefore retain a more adversarial latent-contamination stress test to examine that regime directly.

These latents are sampled from a broader Gaussian distribution that lies away from the main data manifold. Figure 5 visualizes the generated samples projected onto the first two principal components. Standard Flow Matching ($\gamma = 0$, Left) is more easily distracted by these contaminants: the learned flow attempts to fit both inlier and adversarial target signals, resulting in a distorted latent structure. In contrast, $\gamma$-Flow Matching ($\gamma = 0.5$, Right) downweights the contaminants via $\hat{p}_t^\gamma$, thereby preserving closer alignment with the dominant inlier manifold. This stress test therefore illustrates the regime in which the proposed weighting is most helpful: not necessarily already-clean targets, but target geometries that are distorted by contamination or poor support.

Finally, we examine the relationship between vector field smoothness and the number of function evaluations (NFE) required by an ODE solver. We consider a range of NFE values and measure the Fréchet distance between generated images and real images in the latent space. In our experiments, smoother learned vector fields were often associated with better quality at lower NFE. This should be interpreted as an empirical tendency in the tested setting rather than as a universal consequence of the method. In contrast, with a rough vector field (standard FM), reducing the step size (increasing NFE) does not necessarily improve sample quality: the solver is approximating a poor flow that overfits the voids. High NFE efficiency is often

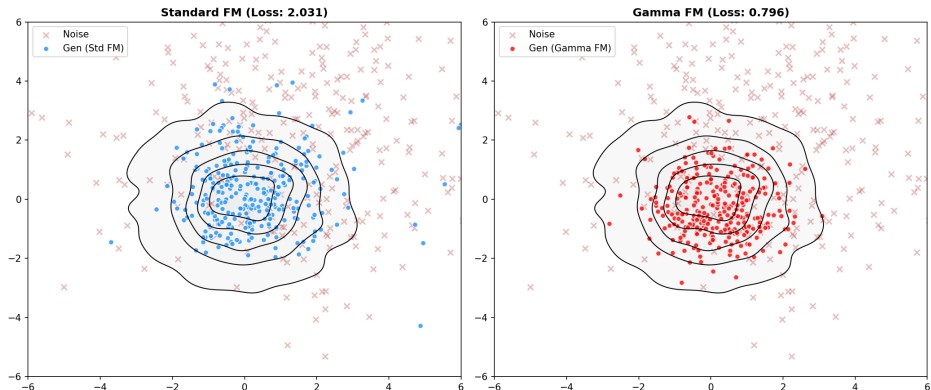

Figure 5: Stress test under a more adversarial latent-contamination pattern. Left: Standard FM ($\gamma = 0$) is more easily misled by adversarial latents and learns a distorted manifold. Right: $\gamma$-FM ($\gamma = 0.5$) downweights the contaminants via $\hat{p}_t^{\gamma}$ and better preserves the dominant inlier structure. This figure is intended to highlight the regime in which density weighting is most useful. In a separate clean-target rerun, where the target geometry was not distorted by contamination, the same weighting mechanism remained active but did not yield a comparably clear gain in decoded-image quality.

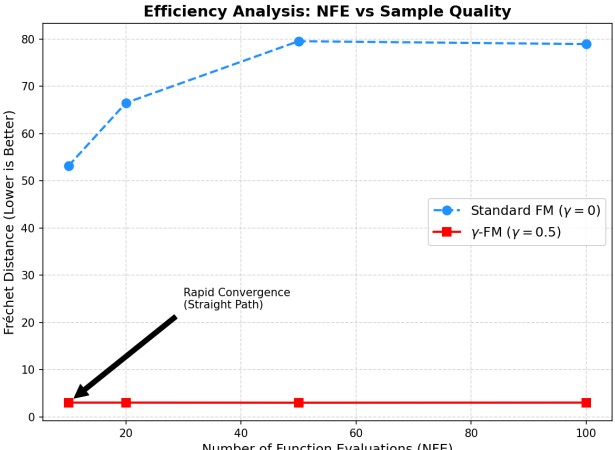

Figure 6: Efficiency Analysis (NFE vs. quality). Comparison of generation quality across different ODE solver steps in the tested latent-CIFAR setting. Red ($\gamma$-FM): Reaches near-optimal quality at relatively low NFE, consistent with a smoother and better-conditioned flow in this experiment. Blue (Standard FM): Shows diminishing returns as NFE increases, suggesting that finer ODE integration does not compensate for a less well-conditioned vector field–a tendency we refer to informally as an Inverse Precision Paradox. This behavior is an empirical observation in the present setting and should not be interpreted as a universal consequence of standard flow matching.

associated with flow rectification methods that enforce straight trajectories (Liu et al., 2023). Our results suggest that $\gamma$-weighting achieves a similar efficiency gain by smoothing the vector field in void regions, effectively removing the stiff components of the dynamics.

## 5   Conclusion

We introduced $\gamma$-Flow Matching as a density-weighted alternative to standard flow matching for high-dimensional generative flows. The presentation separates what is formally established from what is offered as geometric or spectral intuition: the weighted objective suppresses low-density variance contributions, the

porous-medium analogy clarifies a "void rejection" mechanism, and the weighted Dirichlet viewpoint suggests a route toward smoother learned vector fields. Empirically, the evaluation protocol is designed to test these claims under a shared-time density-estimation scheme, with explicit comparison to FM+Jac and with decoded-image metrics that quantify both fidelity and diversity.

Looking forward, the implications of this geometric framework extend well beyond image synthesis. One promising direction is the theoretical expansion into Optimal Transport, where the $\gamma$-weighted kinetic energy suggests a new class of transport costs that naturally penalize paths through low-density regions. Furthermore, the principle of void rejection holds significant potential for Causal Flow Matching; by automatically downweighting regions with poor support (i.e., violations of the positivity assumption), $\gamma$-FM could enable more robust estimation of counterfactuals and interventional distributions in high-dimensional causal discovery. We conclude that density-weighted regression offers a principled path for organizing learning on the data manifold, providing a robust foundation for next-generation generative and causal modeling.

As with many efficiency-improving advances in generative modeling, the proposed method may support beneficial uses such as simulation and data augmentation, but it may also lower the cost of producing synthetic content at scale. The present paper is intended as a methodological contribution, and we do not claim application-specific safeguards beyond standard responsible use and evaluation in controlled research settings.

## Code availability

The Python notebooks used to reproduce the simulation studies are available at [https://github.com/shinto-eguchi/gamma-flow-matching](https://github.com/shinto-eguchi/gamma-flow-matching).

**OpenReview.** The review page for this paper is available at [https://openreview.net/forum?id=LBlkVBDRdu](https://openreview.net/forum?id=LBlkVBDRdu).

## Acknowledgements

The author is grateful to the Action Editor and the anonymous reviewers for their careful reading, constructive comments, and encouraging assessment of the work. Their feedback helped improve the clarity, positioning, and presentation of the paper. The author also thanks the TMLR editorial team for coordinating the review process.

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

## A  $\gamma$-Stein connections and geodesic viewpoint

We briefly sketch how the $\gamma$-divergence induces a Riemannian structure and an affine connection which are naturally expressed in terms of $\gamma$-Stein operators. Let $\{q_\theta\}_{\theta \in \Theta}$ be a parametric family of densities with score functions

$$s_i(x;\theta) = \partial_{\theta^i} \log q_\theta(x)\,.$$

The $\gamma$-divergence $D_\gamma(p,q)$ generates a Riemannian metric $g^{(\gamma)}$ and a pair of dual affine connections $(\nabla^{(\gamma)}, \nabla^{(\gamma)*})$ on the statistical manifold $\mathcal{M} = \{q_\theta\}$ according to the theory of minimum contrast geometry (Eguchi, 1992; 2009). Ignoring an irrelevant constant factor, the induced metric takes the form

$$g_{ij}^{(\gamma)}(\theta) = \int s_i(x;\theta)\, s_j(x;\theta)\, q_\theta(x)^{1+\gamma}\, dx\,,$$

which coincides with the $\gamma$-weighted Fisher information. Equivalently, if we define the escort measure

$$d\mu_{\theta,\gamma}(x) = \frac{q_\theta(x)^{1+\gamma}}{Z_\gamma(\theta)}\, dx\,,$$

then

$$g_{ij}^{(\gamma)}(\theta) = Z_\gamma(\theta)\, \mathbb{E}_{\mu_{\theta,\gamma}}\big[s_i s_j\big]\,.$$

For a tangent vector $\xi = \xi^i \partial_i \in T_\theta \mathcal{M}$, we associate the score field

$$u_\xi(x;\theta) = \xi^i s_i(x;\theta)\,.$$

The map $\xi \mapsto u_\xi$ embeds the tangent space into the weighted Hilbert space

$$\mathcal{H}_{\theta,\gamma} = L^2\big(q_\theta^{1+\gamma}(x)\, dx\big)\,,$$

and the metric can be written as

$$g_\theta^{(\gamma)}(\xi,\eta) = \langle u_\xi, u_\eta \rangle_{\theta,\gamma} = \int u_\xi(x;\theta) u_\eta(x;\theta)\, q_\theta(x)^{1+\gamma} dx\,.$$

Following the general construction of Eguchi (1992), the affine connection $\nabla^{(\gamma)}$ can be realized as the $L^2$-projection of the $\theta$-derivative of score fields back onto the score span. More precisely, for each coordinate direction $\partial_k$ we consider $\partial_k s_i = \partial_{\theta^k} \partial_{\theta^i} \log q_\theta$ as an element of $\mathcal{H}_{\theta,\gamma}$ and define the projection

$$\Pi_{\theta,\gamma}\big[\partial_k s_i\big] = \Gamma_{ik}^{(\gamma)\,j}(\theta) s_j(\cdot;\theta)\,,$$

where the connection coefficients $\Gamma_{ik}^{(\gamma)\,j}$ are determined by the orthogonality relation

$$\left\langle \partial_k s_i - \Gamma_{ik}^{(\gamma)\,j} s_j,\ s_\ell \right\rangle_{\theta,\gamma} = 0\,, \qquad \forall\,\ell\,.$$

This yields the explicit formula

$$g_{j\ell}^{(\gamma)}(\theta)\,\Gamma_{ik}^{(\gamma)\,j}(\theta) = \int \partial_k s_i(x;\theta)\, s_\ell(x;\theta)\, q_\theta(x)^{1+\gamma}\, dx\,,$$

or equivalently,

$$\Gamma_{ik}^{(\gamma)\,j}(\theta) = g^{(\gamma)\,j\ell}(\theta)\,\int \partial_k s_i(x;\theta)\, s_\ell(x;\theta)\, q_\theta(x)^{1+\gamma}\, dx\,.$$

The $\gamma$-Stein operator

$$\mathcal{A}_{q_\theta}^{(\gamma)} f = q_\theta^{-1}\,\nabla\!\cdot\!\big(q_\theta^{\gamma+1} f\big)$$

encodes a weighted divergence with respect to the escort measure $\mu_{\theta,\gamma}$. The Stein identity $\mathbb{E}_{q_\theta}[\mathcal{A}_{q_\theta}^{(\gamma)} f] = 0$ can be interpreted as an orthogonality condition in $\mathcal{H}_{\theta,\gamma}$, and integration by parts allows one to rewrite the right-hand side of the above expression for $\Gamma^{(\gamma)}$ in terms of $\gamma$-Steinized moments of the score fields. Thus the connection $\nabla^{(\gamma)}$ may be viewed as a $\gamma$-Stein connection associated with the escort family $\{\mu_{\theta,\gamma}\}$.

Finally, a smooth curve $\theta(t)$ is a $\nabla^{(\gamma)}$-geodesic if and only if it satisfies

$$\ddot{\theta}^i(t) + \Gamma_{jk}^{(\gamma)\,i}(\theta(t))\dot{\theta}^j(t)\dot{\theta}^k(t) = 0\,.$$

In terms of score fields, this condition is equivalent to requiring that the associated field

$$u_t(x) = \dot{\theta}^i(t)s_i(x;\theta(t))$$

evolves in $\mathcal{H}_{\theta,\gamma}$ according to

$$\Pi_{\theta(t),\gamma}\big[\partial_t u_t\big] = 0\,,$$

that is, $u_t$ is covariantly constant under the $\gamma$-Stein connection along the curve. This provides a geometric counterpart to the sample-space flow governed by the nonlinear Fokker–Planck equation, and suggests a dual picture in which $\gamma$-flow matching approximately follows geodesics on the statistical manifold endowed with the $\gamma$-Fisher metric and the $\gamma$-Stein connection. A more detailed study of this escort geometry is left for future work; see, e.g., Eguchi & Kato (2010); Matsuzoe (2017) for related developments on escort distributions.

## B  Hyperparameter and proxy sensitivity

A potential concern with density-weighted regression is the computational overhead introduced by the $k$-Nearest Neighbors ($k$-NN) estimation within each training batch. To address this, we first evaluate the computational cost and stability of the practical $k$-NN surrogate used in our implementation. We then examine how the surrogate behaves as the minibatch size $B$ and the ambient dimension $D$ vary under the shared-time protocol.

The purpose of this appendix is operational rather than asymptotic: we do not claim a universal high-dimensional guarantee for $k$-NN density estimation. Instead, we document how the within-batch geometry and the induced practical weights behave in the regime relevant to our experiments.

## C  Computational cost of the $k$-NN surrogate

We first measure the wall-clock time per training iteration while varying the number of neighbors $k \in \{5, 10, 20, 50, 100\}$. All experiments use batch size $B = 512$ and are executed on a single NVIDIA T4 GPU. The reported values are averaged over 1000 iterations.

**Results.** Table 2 shows that the computational cost is effectively invariant to the choice of $k$. The average time per iteration remains approximately 2.0–2.5 ms across all tested values. This occurs because the dominant cost arises from the pairwise distance computation ($O(B^2)$), which is highly parallelized on modern GPUs. The additional top-$k$ selection step introduces negligible overhead for typical batch sizes. The loss values also show that training stability is insensitive to the precise choice of $k$. Therefore, the practical surrogate adds minimal computational burden compared with standard Flow Matching.

Table 2: Ablation study on neighbor size $k$. Time per iteration remains nearly constant.

| $k$ | 5 | 10 | 20 | 50 | 100 |
|---|---|---|---|---|---|
| Time (ms/iter) | 2.1 | 2.1 | 2.2 | 2.1 | 2.1 |
| Avg Loss | 1.62 | 1.60 | 1.66 | 1.57 | 1.65 |

### C.1 Sensitivity to batch size and ambient dimension

To better understand the geometric behavior of the practical surrogate, we conducted a diagnostic experiment varying both the minibatch size $B$ and the effective ambient dimension $D$ under the shared-time protocol.

The analysis focuses on three quantities:

- the $k$-NN distance statistic $d_k(x)$,

- the inverse-distance proxy $\rho_k(x) = 1/(d_k(x) + \varepsilon)$,

- the induced practical weights $w(x)$ used in the $\gamma$-weighted regression objective.

**Dimension sensitivity.** At fixed batch size $B = 512$, the raw $k$-NN geometry becomes increasingly concentrated as the ambient dimension increases. The coefficient of variation of the mean-$k$ distance decreases from $\mathrm{CV}(d_k) = 0.087$ at $D = 32$ to $0.044$ at $D = 256$. Similarly, the contrast ratio $q_{0.8}(d_k)/q_{0.2}(d_k)$ decreases from 1.158 to 1.078. The same trend appears for the proxy $\rho_k$, where $\mathrm{CV}(\rho_k)$ decreases from 0.086 to 0.045. These results are consistent with the well-known concentration of pairwise distances in higher dimensions.

**Batch-size sensitivity.** At fixed dimension $D = 256$, increasing the minibatch size from $B = 128$ to $B = 1024$ slightly strengthens the within-batch ordering signal. The coefficient of variation $\mathrm{CV}(d_k)$ increases from 0.041 to 0.048, and the contrast ratio $q_{0.8}(d_k)/q_{0.2}(d_k)$ increases from 1.071 to 1.085. Thus, larger batches provide a modestly sharper local ordering signal.

**Behavior of the practical weights.** Importantly, the final weights are obtained through a rank-based monotone transform of the proxy rather than from its absolute scale. Consequently, summary statistics of the induced weights, such as $\mathrm{ESS}(w)/B$, remain nearly invariant to the ambient dimension. This behavior is by design: the practical surrogate is intended to provide a stable within-batch ordering signal rather than to estimate densities on an absolute scale. In practice, the surrogate preserves sufficient ordering information even though the raw $k$-NN geometry becomes less contrasted in higher dimensions.

Figure 7 visualizes these trends. The results show that although the raw geometric contrast weakens as dimension increases, the rank-based practical surrogate remains operationally stable for the range of $B$ and $D$ used in our experiments.

## D  Latent-flow modelling of CIFAR-10

**Implementation details.** The latent CIFAR-10 experiments use a robust-target setup with $n_{\text{train}} = 10000$ images and contamination fraction 0.40, consisting of 6000 inlier-class images and 4000 outlier-class images. All latent-space experiments use the same frozen autoencoder and the same decoding pipeline across FM, shared-time $\gamma$-FM, and FM+Jac. The autoencoder is a pre-trained EMA checkpoint of a `CleanAE` architecture with base width 64 and latent dimension $d_z = 256$.

Table 3: Sensitivity of the $k$-NN geometry and the induced practical weights to batch size $B$ and ambient dimension $D$.

| Setting | $B$ | $D$ | $\mathrm{CV}(d_k)$ | $q_{0.8}(d_k)/q_{0.2}(d_k)$ | $\mathrm{CV}(\rho_k)$ | $\mathrm{ESS}(w)/B$ |
|---|---|---|---|---|---|---|
| | | | Fixed $D = 256$ (varying $B$) | | | |
| | 128 | 256 | 0.0407 | 1.071 | 0.0412 | 0.603 |
| | 256 | 256 | 0.0414 | 1.073 | 0.0421 | 0.603 |
| | 512 | 256 | 0.0443 | 1.078 | 0.0454 | 0.603 |
| | 1024 | 256 | 0.0481 | 1.085 | 0.0499 | 0.603 |
| | | | Fixed $B = 512$ (varying $D$) | | | |
| | 512 | 32 | 0.0873 | 1.158 | 0.0864 | 0.603 |
| | 512 | 64 | 0.0668 | 1.119 | 0.0671 | 0.603 |
| | 512 | 128 | 0.0515 | 1.091 | 0.0522 | 0.603 |
| | 512 | 256 | 0.0445 | 1.078 | 0.0455 | 0.603 |

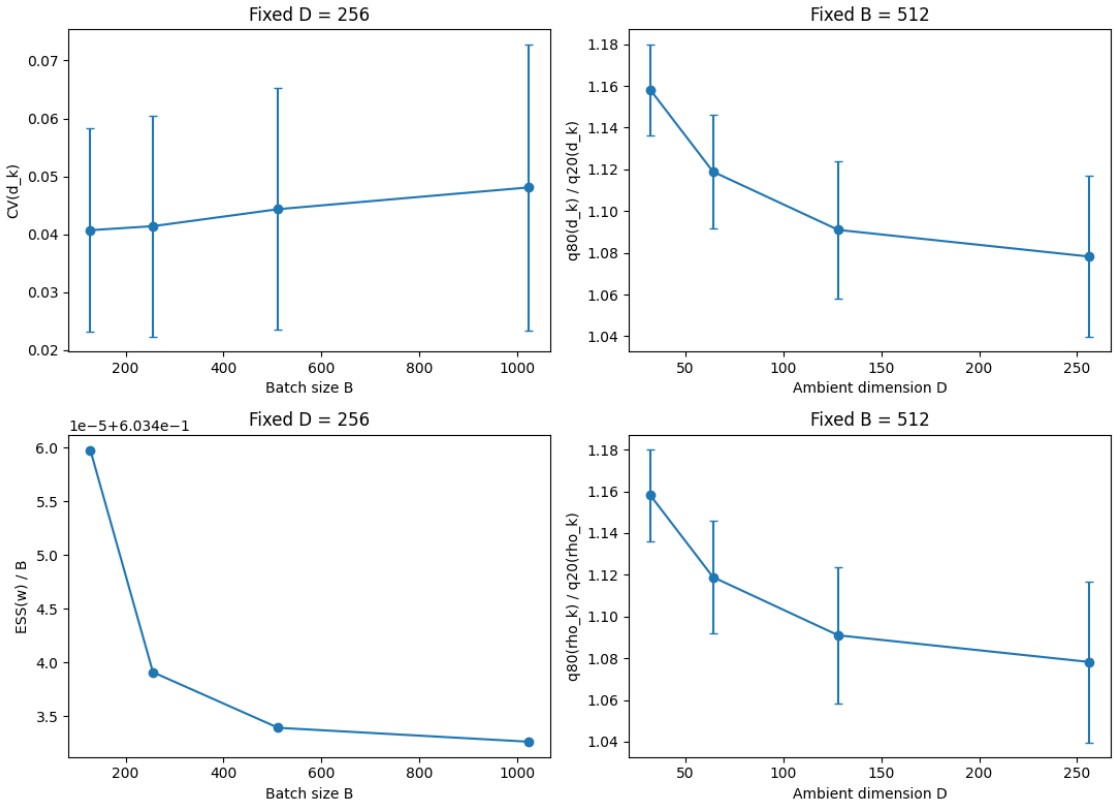

Figure 7: Operational sensitivity of the practical $k$-NN surrogate. Top: behavior of the raw $k$-NN distance statistic. Bottom: behavior of the induced practical weights and proxy.

The latent flow backbone is a `VelocityMLP` with hidden width 512 and depth 3, trained for 60 epochs with batch size 512 using Adam with learning rate $10^{-3}$ and EMA decay 0.999. In the shared-time protocol, each minibatch draws a single $t \sim \mathcal{U}[0,1]$ and constructs all particles at that common time. The practical weighting uses a rank-based monotone $k$-NN surrogate with $k = 10$, lower and upper rank quantiles 0.20 and 0.80, rank scale 2.0, and weight clipping to $[10^{-3}, 5 \times 10^1]$.

For FM+Jac, the penalty coefficient is $\lambda_J = 10^{-3}$, and the Jacobian norm is estimated by a single Hutchinson probe with Rademacher noise during both training and evaluation. Sampling uses a fixed-step midpoint integrator with 100 steps. Decoding uses retrieval-skip with a skip bank of 600 encoded training images,

$k = 2$, and $\tau = 0.5$. Latent MMD is computed with subsample size 2000, and smoothness is evaluated with 2000 random latent samples and one Rademacher probe. FID, Recall, and Coverage are computed from 5000 generated images, with metric batch size 128 and coverage parameter $k = 5$; the reference set consists of the 1000 test images of the inlier class. Multi-seed runs use seeds 42, 1042, and 2042 on a single NVIDIA Tesla T4 GPU.

This appendix provides additional visual results for the latent-flow CIFAR-10 experiments in Section 4.2. We make the experimental protocol explicit by reporting the autoencoder architecture, latent dimensionality, flow backbone, shared-time minibatch construction, practical $k$-NN surrogate parameters, Jacobian-penalty implementation, fixed-step sampling protocol, decoding protocol, the number of generated samples used for FID/Recall/Coverage, random seeds, and hardware.

Subsequently, we trained the flow matching models within this fixed latent space. Figure 8 shows representative decoded samples obtained from FM, $\gamma$-FM (shared-$t$), and FM+Jac. The decoding process uses retrieval-skip with parameters $k = 2$ and $\tau = 0.5$. In the comparison, the same frozen decoder and the same retrieval-skip settings are used for FM, $\gamma$-FM (shared-$t$), and FM+Jac so that image-space differences reflect changes in the latent-flow objective rather than changes in the representation or decoding pipeline. These samples are intended as a qualitative complement to the quantitative metrics, showing that the overall decoded sample quality remains reasonable across the compared methods under a fixed latent representation.

### D.1 Minibatch-size sensitivity of the shared-time weighting proxy

A natural concern is whether the shared-time $k$-NN proxy used in $\gamma$-FM becomes unreliable when the minibatch is small. To examine this point, we repeated the latent CIFAR-10 experiment under the same latent representation, frozen autoencoder, decoding pipeline, optimizer, and training budget as in the main experiment, varying only the minibatch size

$$B \in \{128, 256, 512, 1024\}.$$

We compared shared-time FM and shared-time $\gamma$-FM with $\gamma = 1$, and report means and standard deviations over three random seeds in Table 4.

The main observation is that the weighting proxy did not collapse at smaller batch sizes in this regime. For $\gamma$-FM, the final effective sample size ratio ESS/$B$ remained between 0.512 and 0.526 across all tested values of $B$, while the final weight quantile ratio $q_{0.95}/q_{0.05}$ stayed near 20.09 throughout. Thus, the proxy retained a stable, nontrivial dynamic range rather than degenerating as $B$ decreased.

As expected, image-space quality improved with minibatch size for both FM and $\gamma$-FM. Hence, the degradation at small $B$ should not be interpreted as a pathology specific to density weighting; rather, it reflects the general difficulty of minibatch training in this contaminated latent-generation setting. Within this shared trend, the most consistent advantage of $\gamma$-FM appears in the geometry of the learned field: at low to moderate batch sizes ($B = 128, 256, 512$), the smoothness proxy is lower for $\gamma$-FM than for FM, while recall and coverage remain broadly comparable. This is consistent with the intended role of the weighting scheme as an implicit geometric regularizer.

As a complementary sanity check, we also repeated the same minibatch-size experiment under a clean-target latent protocol, that is, without outlier contamination in the training targets. In this rerun, the weighting mechanism remained clearly active, with final ESS/$B$ near 0.52 and the final weight quantile ratio near 20 across batch sizes. However, the decoded-image advantage largely disappeared. At $B = 1024$, for example, shared-time FM achieved slightly better FID, Recall, and Coverage than shared-time $\gamma$-FM, while the smoothness values became nearly identical. This clean-target rerun is therefore included mainly as a sanity check on the mechanism rather than as a regime in which $\gamma$-FM is expected to dominate FM.

FM_sharedt

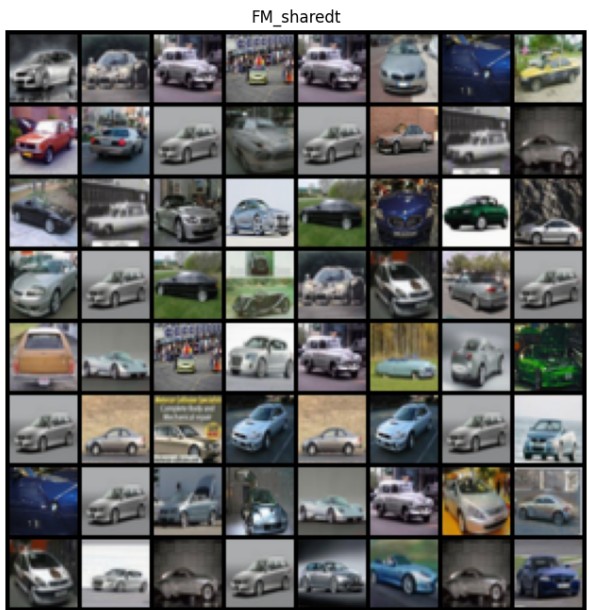

gFM_sharedt_sweep_g0p5_seed2

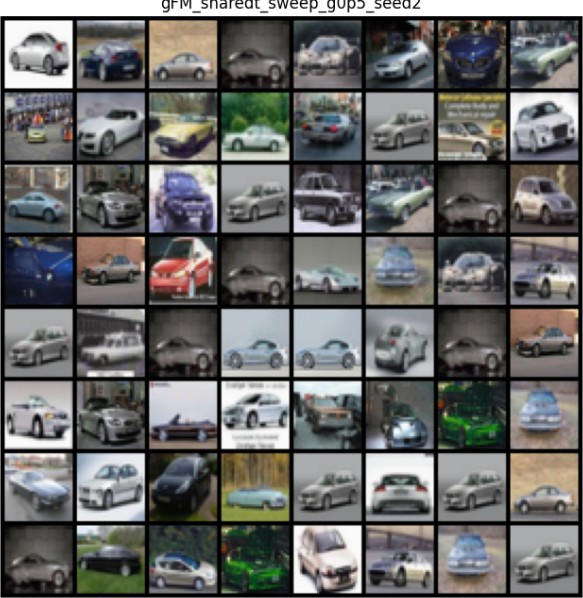

(a) FM (shared-$t$) samples (retrieval-skip: $k = 2$, $\tau = 0.5$).

(b) $\gamma$-FM with $\gamma = 0.5$ (retrieval-skip: $k = 2$, $\tau = 0.5$).

gFM_sharedt_g1p0_seed2

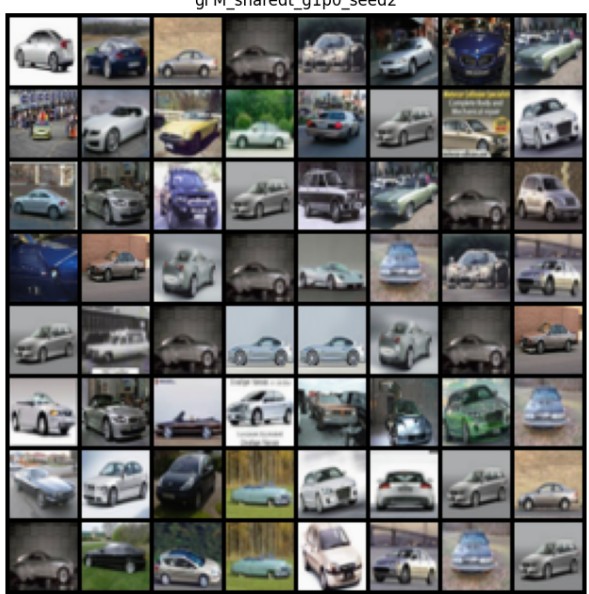

FM_Jac_sharedt_lam0p001_seed2

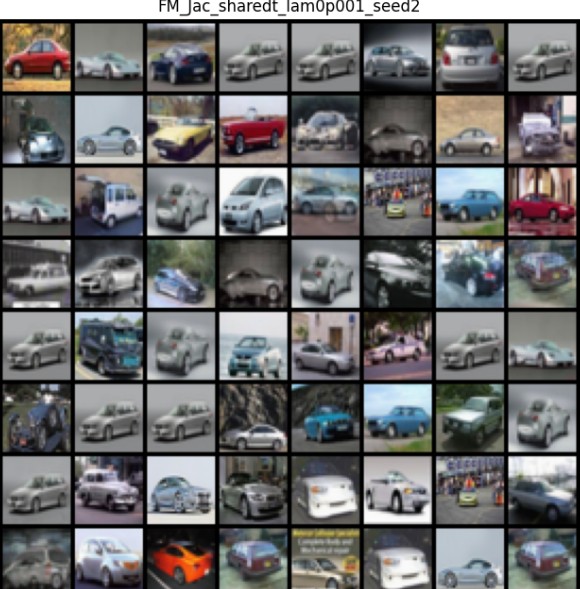

(c) $\gamma$-FM with $\gamma = 1.0$ (retrieval-skip: $k = 2$, $\tau = 0.5$).

(d) FM+Jac with $\lambda_J = 10^{-3}$ (retrieval-skip: $k = 2$, $\tau = 0.5$).

Figure 8: Representative decoded samples for FM (shared-$t$), $\gamma$-FM ($\gamma = 0.5, 1.0$), and FM+Jac under the latent CIFAR-10 protocol. All methods use the same frozen decoder and the same retrieval-skip settings ($k = 2$, $\tau = 0.5$). This figure is intended as a qualitative complement to the quantitative comparisons in Table 1.

Table 4: Minibatch-size ablation for shared-time FM and shared-time $\gamma$-FM ($\gamma = 1$) on latent CIFAR-10 under the robust-target protocol. Each entry is the mean $\pm$ standard deviation over three random seeds. Lower FID and smoothness are better, while higher Recall and Coverage are better. For FM, ESS$/B = 1$ by construction because the weights are uniform.

| Method | $B$ | FID$\downarrow$ | Recall$\uparrow$ | Coverage$\uparrow$ | Smoothness$\downarrow$ | ESS$/B$ | $q_{0.95}/q_{0.05}$ |
|---|---|---|---|---|---|---|---|
| FM (shared-$t$) | 128 | $64.321 \pm 0.757$ | $0.370 \pm 0.012$ | $0.893 \pm 0.012$ | $64.813 \pm 3.335$ | 1.000 | 1.00 |
| FM (shared-$t$) | 256 | $62.407 \pm 2.303$ | $0.402 \pm 0.012$ | $0.907 \pm 0.005$ | $33.768 \pm 4.525$ | 1.000 | 1.00 |
| FM (shared-$t$) | 512 | $59.063 \pm 0.759$ | $0.388 \pm 0.008$ | $0.915 \pm 0.006$ | $9.136 \pm 0.110$ | 1.000 | 1.00 |
| FM (shared-$t$) | 1024 | $53.054 \pm 0.965$ | $0.406 \pm 0.020$ | $0.938 \pm 0.004$ | $1.209 \pm 0.175$ | 1.000 | 1.00 |
| $\gamma$-FM ($\gamma = 1$) | 128 | $66.076 \pm 0.657$ | $0.372 \pm 0.010$ | $0.889 \pm 0.011$ | $50.795 \pm 2.236$ | 0.512 | 20.09 |
| $\gamma$-FM ($\gamma = 1$) | 256 | $64.339 \pm 1.434$ | $0.384 \pm 0.017$ | $0.877 \pm 0.015$ | $27.697 \pm 2.384$ | 0.518 | 20.09 |
| $\gamma$-FM ($\gamma = 1$) | 512 | $61.198 \pm 0.735$ | $0.395 \pm 0.009$ | $0.915 \pm 0.011$ | $8.788 \pm 0.194$ | 0.524 | 20.09 |
| $\gamma$-FM ($\gamma = 1$) | 1024 | $54.223 \pm 0.486$ | $0.412 \pm 0.019$ | $0.933 \pm 0.005$ | $1.253 \pm 0.155$ | 0.526 | 20.09 |

